# Distinct roles of cortical layer 5 subtypes in associative learning

Sara Moberg[1,2], Michele Garibbo[3,4,12], Camille Mazo [5,12], Ariel Gilad [6], Dietmar Schmitz[1,7,8,9,10,11], Rui Ponte Costa [3], Matthew E. Larkum [2] ✉ & Naoya Takahashi [5] ✉

Adaptive behavior relies on associating sensory cues with rewarding or aversive outcomes. In mammals, the primary sensory cortex processes stimuli and distributes information to cortical and subcortical targets. Layer 5 (L5) contains two major projection neuron classes, intratelencephalic (IT) and extratelencephalic (ET); however, their roles in associative learning remain unclear. Using transgenic mice, we identified IT and ET neurons in primary somatosensory cortex and tracked their activity with longitudinal two-photon imaging during Pavlovian conditioning with whisker stimulation. IT neurons stably encoded stimulus identity across training, whereas ET neurons showed dynamic changes that paralleled the emergence of anticipatory licking. Chemogenetic silencing of each subtype impaired learning in distinct, phase-specific ways. A reinforcement-learning model reproduced these dynamics, suggesting that IT neurons provide stable sensory representations needed to form cue-reward associations, while ET neurons encode reward expectation to refine behavior. These findings reveal complementary, cell-type-specific contributions of L5 neurons to associative learning.

Associative learning is essential for selecting appropriate behaviors by linking stimuli to different outcomes. The learning process consists of multiple components, including acquisition and refinement, and involves diverse neural circuits across the brain that function in concert[1,2]. Primary sensory cortical areas are crucial for first processing and analyzing sensory stimuli, but also for linking these stimuli to behaviorally relevant outcomes, such as rewards or threats[3–6]. During stimulus-reward associative learning, neurons in these areas alter their response patterns to stimuli based on their relevance to rewards[3,7–9].

These learning-associated changes in neuronal activity are thought to be transmitted to other brain regions outside the sensory cortices. While recent studies have begun to elucidate the contribution to learning of cortical neurons projecting to distant brain regions[5], little is known about the activity changes these projection neurons undergo during learning and their functional roles throughout the learning process.

Layer 5 (L5) serves as the main output layer of sensory cortices and consists of two major groups of outward projecting pyramidal

[1]Charité - Universitätsmedizin Berlin, corporate member of Freie Universität Berlin and Humboldt Universität zu Berlin, Einstein Center for Neurosciences Berlin, Berlin, Germany. [2]Institute for Biology, Humboldt University of Berlin, Berlin, Germany. [3]Centre for Neural Circuits and Behaviour, Department of Physiology, Anatomy and Genetics, University of Oxford, Oxford, United Kingdom. [4]Bristol Computational Neuroscience Unit, Intelligent Systems Lab, Faculty of Engineering, University of Bristol, Bristol, United Kingdom. [5]University of Bordeaux, CNRS, Interdisciplinary Institute for Neuroscience, IINS, UMR 5297, Bordeaux, France. [6]Department of Medical Neurobiology, Faculty of Medicine, Institute for Medical Research Israel-Canada (IMRIC), The Hebrew University of Jerusalem, Jerusalem, Israel. [7]Charité - Universitätsmedizin Berlin, corporate member of Freie Universität Berlin and Humboldt Universität zu Berlin, Neuroscience Research Center, Berlin, Germany. [8]Charité - Universitätsmedizin Berlin, corporate member of Freie Universität Berlin and Humboldt Universität zu Berlin, NeuroCure Cluster of Excellence, Berlin, Germany. [9]Bernstein Center for Computational Neuroscience Berlin, Berlin, Germany. [10]German Center for Neurodegenerative Diseases (DZNE) Berlin, Berlin, Germany. [11]Max-Delbrück Center for Molecular Medicine in the Helmholtz Association, Berlin, Germany. [12]These authors contributed equally: Michele Garibbo, Camille Mazo. ✉e-mail: matthew.larkum@hu-berlin.de; naoya.takahashi@u-bordeaux.fr

neurons, intratelencephalic (IT) and extratelencephalic (ET) neurons. IT neurons project bilaterally to other cortical areas and the striatum, while ET neurons project predominantly to subcortical areas in the ipsilateral hemisphere, including the striatum, higher-order thalamic nuclei, midbrain and pons[10–12]. Recent advances in transgenic[13] and retrograde-viral targeting[14] technology have begun to shed light on the distinct contribution of each projection neuronal subtype to cortical function and information processing, such as sensory detection and discrimination[4,5,15,16], decision making[17], and movement control[18–21]. But as yet, the question still remains as to how IT and ET neurons contribute to learning.

The primary somatosensory cortex (S1) has been shown to engage in vibrotactile discrimination in humans[22], primates[23], and rodents[24]. In the present study, we sought to determine how the activity patterns of S1 L5 projection neuronal subtypes evolve during learning of vibrotactile discrimination, specifically in the context of stimulus-reward associations. To study the learning process, we designed an appetitive Pavlovian conditioning task, where head-restrained mice were trained to distinguish between two vibratory stimuli delivered to their whiskers with different frequencies, with one stimulus paired with a water reward and the other not. By leveraging transgenic mouse lines, we selectively targeted and tracked the activity of either L5 IT or ET neuronal populations in S1 during the learning process. Using in vivo two-photon imaging, we measured calcium activity in the apical dendritic trunks, which highly correlates with somatic activity of L5 pyramidal neurons[25,26], and found distinct response patterns between IT and ET neurons to stimuli, along with changes in these patterns during learning. By chemogenetically silencing selective L5 subtypes, we found that both IT and ET neurons are critical for learning, with each

subtype contributing to distinct aspects of the learning process. To further elucidate the specific roles of IT and ET neurons, based on their response properties, we developed a theoretical model inspired by classical reinforcement learning frameworks. The model suggests that IT neurons maintain internal representations of the sensory world that initiate the formation of stimulus-reward associations, whereas ET neurons contribute to the refinement of learned responses, enhancing discriminability between rewarding and non-rewarding stimuli.

## Results

### Whisker stimulus-reward associative learning depends on S1

To study reward-based associative learning in mice, we developed a Pavlovian trace conditioning task in which head-restrained mice were trained to associate one of two stimuli, i.e., a conditioned stimulus (CS+) vs. a non-conditioned stimulus (CS−), with water reward (Fig. 1a). The CS+ consisted of C2-whisker stimulation at 10 Hz for 1 s, followed by a short delay (trace interval; 0.5 s) before the delivery of a reward. For CS− trials, the same whisker was stimulated at 5 Hz with no reward. Over the course of daily training sessions, the mice gradually learned to discriminate between the two stimuli (Fig. 1b, c). Initially, they increased the frequency of anticipatory licking (i.e., licking from stimulus onset to the time of reward delivery) for both stimuli. With continued training, they developed a selective increase in anticipatory licking for the CS+, while licking to the CS− tended to decrease. The task performance, i.e., CS discriminability, was quantified as the area under a receiver operating characteristic curve (auROC) calculated on anticipatory licks in CS+ vs. CS− trials for each session (Fig. 1d)[27,28]. Within five days, 18 out of 20 mice reached a proficient level (auROC >0.7) to discriminate the stimuli (Fig. 1e).

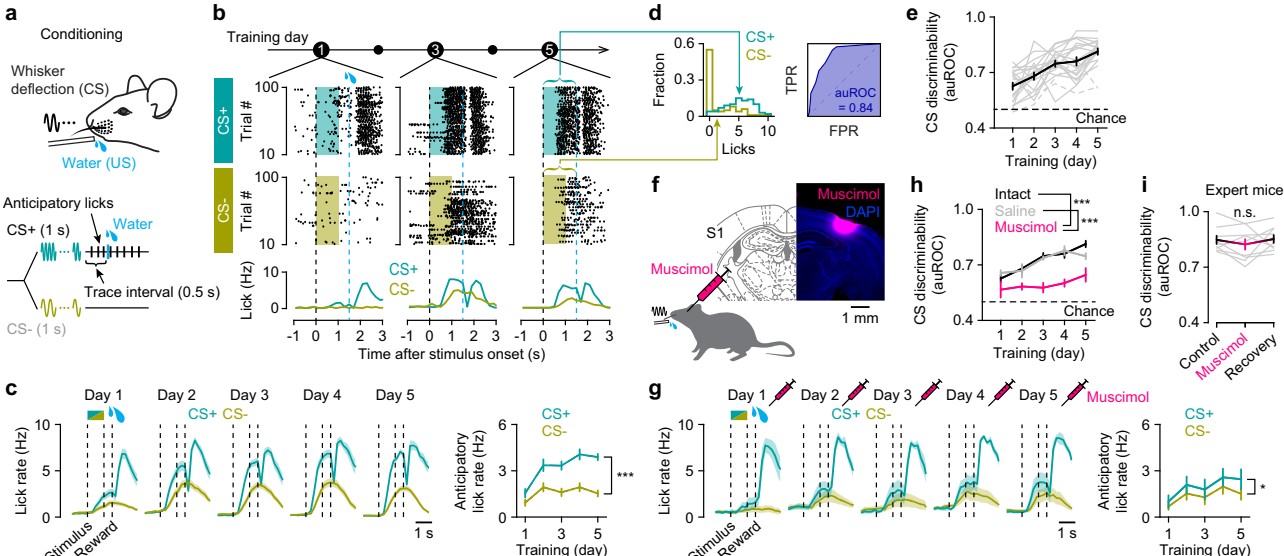

**Fig. 1 | Whisker stimulus-reward associative learning depends on S1.**
**a** Behavioral task design. Mice were exposed to two different whisker stimuli of different frequencies: one stimulus (CS+, 1 s) followed by a reward after a short delay (trace interval, 0.5 s), and the other (CS−, 1 s) left unrewarded. **b** Example training sessions (Days 1, 3, and 5) from a mouse showing anticipatory (from stimulus onset to reward onset) and consummatory (from reward onset) licking during CS+ and CS− trials. Licking activities are shown in raster plots (top) and histograms (bottom). Top, raster plot with trial sorted according to stimulus type (CS+ vs. CS−). Shaded areas, stimulus presentation; black dashed lines, stimulus onset and offset; blue dashed lines, reward onset for CS+ trials. **c** Left, average lick rates for CS+ (blue) and CS− (yellow) trials across mice for Days 1–5 (n = 20 mice). Right, evolution of mean anticipatory lick rates during learning (***P = 7.4 × 10⁻¹², F = 217.48; two-way repeated-measure ANOVA). Dashed lines represent CS onset, offset and reward onset. **d** Left, distribution of the anticipatory lick counts during CS+ and CS− trials of an example session (Day 5 in **b**). Right, the area under the

receiver operating characteristic curve (auROC), scoring behavioral performance. TPR: true positive rate; FPR: false positive rate. **e** Evolution of CS discrimination performance over five days of training (n = 20 mice; P = 8.1 × 10⁻¹², F = 21.32; one-way repeated-measure ANOVA). Gray lines, individual mice; gray dashed lines, mice that did not learn the task. **f** Local pharmacological silencing of S1. Inset, coronal section of S1, showing the diffusion of muscimol injected in the C2 barrel column. **g** Same as (**c**) but for mice injected with muscimol in S1 during learning (*P = 0.024, F = 10.24; two-way repeated-measure ANOVA). Note that the difference in anticipatory lick rates between CS+ and CS− was markedly reduced compared to intact S1 (**c**).
**h** Learning trajectory of mice with intact S1 (n = 20 mice), mice with saline injection (n = 6 mice), and mice with muscimol injection (n = 6 mice; P = 1.2 × 10⁻⁴, F = 12.48; two-way mixed ANOVA with post hoc Tukey–Kramer test). **P < 0.01, ***P < 0.001.
**i** Behavioral performance of expert mice injected with muscimol (n = 9 mice; P = 0.47, F = 0.78; one-way repeated-measure ANOVA). Gray lines, individual mice. Data were presented as mean ± SEM. Source data are provided as a Source Data file.

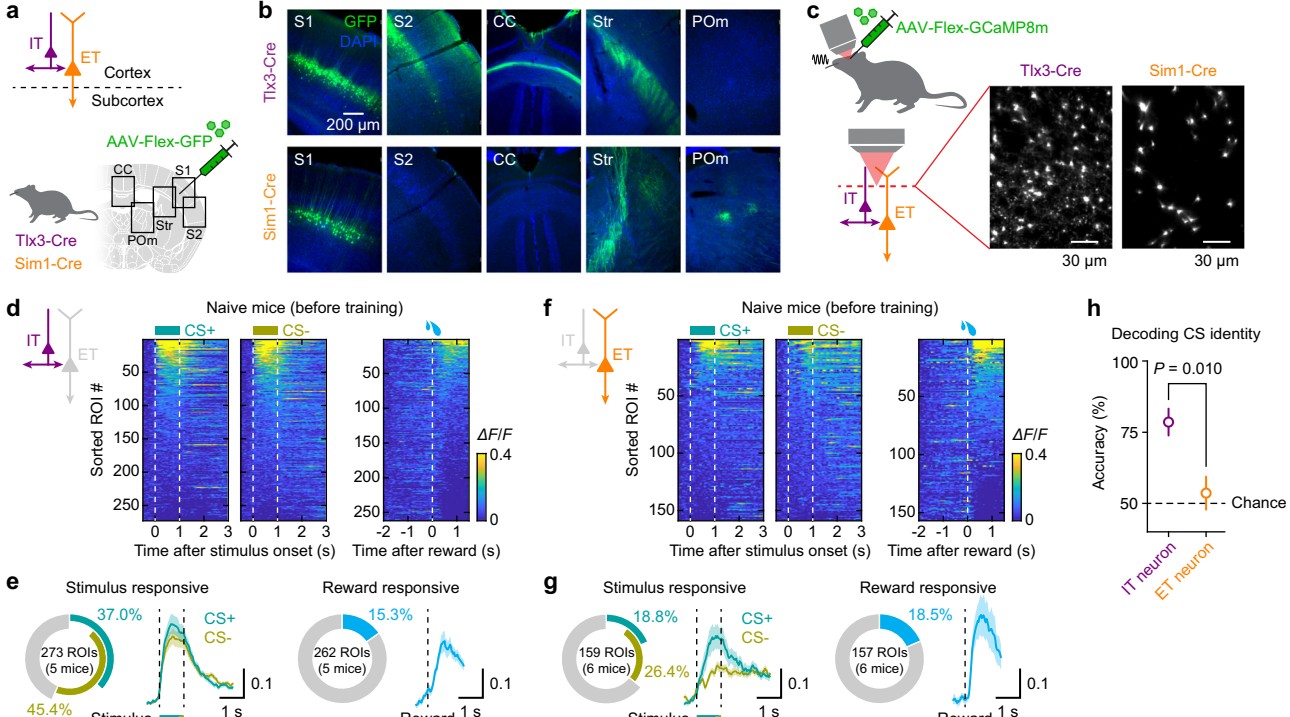

**Fig. 2 | Distinct response patterns of L5 neuronal subtypes to sensory stimuli and reward in naïve mice. a** Schematic showing the genetic and viral intersectional strategy, targeting L5 IT neurons in Tlx3-Cre mice and ET neurons in Sim1-Cre mice. **b** Images from coronal sections from S1 of a Tlx3-Cre mouse (top) and Sim1-Cre mouse (bottom) injected with AAV-Flex-GFP, showing the injection site in S1 and labeled axons in the secondary somatosensory cortex (S2), corpus callosum (CC), striatum (Str), and posterior medial thalamic nucleus (POm). **c** Two-photon calcium imaging from Tlx3-Cre and Sim1-Cre mice injected with AAV-Flex-jGCaMP8m in S1. Inset, example FOVs showing GCaMP-expressing IT neuronal dendrites in a Tlx3-Cre mouse (left) and ET dendrites in a Sim1-Cre mouse (right). **d** Heatmaps of IT neuronal responses to whisker stimuli (left, CS+; middle, CS−) and to water reward (right) in naïve mice ($n = 5$ mice). ROIs in each heatmap are sorted by their mean response amplitudes within 1.5 s of stimulus or reward onset. **e** Left, pie-chart showing the fraction of IT neurons responding to each CS, and their average responses. Dashed line, stimulus onset and offset. Right, pie-chart showing the fraction of IT neurons responding to rewards, and their average responses. Dashed line, reward onset. **f, g** Same as (**d, e**) but for ET neurons ($n = 6$ mice). **h** SVM decoder performance in classifying the CS identity (CS+ or CS−) based on IT neuronal responses (purple; $n = 5$ mice) or ET neuronal responses (orange; $n = 6$ mice; two-sided Student's $t$-test). Data were presented as mean ± SEM. Source data are provided as a Source Data file.

Learning was not specific to the stimulus frequencies, as mice could successfully learn the task with the reversed CS contingency (5 Hz as CS+, 10 Hz as CS−; Fig. S1a).

To test whether our learning paradigm depends on S1, we pharmacologically inactivated it during training. Prior to each training session, muscimol, a GABA$_A$-receptor agonist, was locally injected into the C2 barrel column of S1, contralaterally to the stimulated C2 whisker (Fig. 1f). Silencing S1 impaired learning as mice failed to develop anticipatory licking to the stimuli and poorly distinguished between CS+ and CS−, unlike control mice with intact S1 or saline injection (Fig. 1g, h). In contrast, silencing S1 in expert mice did not affect the behavioral performance, regardless of the CS contingency (Fig. 1i and Fig. S1b, c). Together, these results demonstrate that S1 activity facilitates learning of the task, but once the task is learned, S1 is not necessary for the expression of conditioned responses.

## Baseline differences in stimulus and reward encoding between IT and ET neurons

To investigate the responses of S1 neurons to stimuli and rewards, and how they are modulated during learning, we used in vivo two-photon calcium imaging to monitor neuronal activity over time before, during and after learning. We focused on the two main long-range projection neurons in L5 of S1, IT and ET neurons. Two Cre-driver transgenic mouse lines, Tlx3-Cre and Sim1-Cre, were used to distinguish between IT and ET neurons, respectively (Fig. 2a)[13]. A Cre-dependent viral expression strategy yielded selective labeling of L5

IT neurons in Tlx3-Cre mice and ET neurons in Sim1-Cre mice, as previously reported (Fig. 2b)[13,16,19,20].

Using these mouse lines, we expressed a genetically-encoded calcium indicator, jGCaMP8m[29], in IT and ET neurons. To facilitate stable longitudinal recordings, we imaged apical dendritic trunks as a proxy for global activity as activities in apical trunks and somata have been shown to be highly correlated[25,26]. Imaging was performed from the same field of view (FOV) within the C2 barrel column across multiple days: a baseline session prior to training to assess stimulus- and reward-evoked responses independently, followed by five consecutive sessions during training. On average, the FOV (178.3 × 178.3 µm²) contained 56 IT neurons (34−83 neurons, $n = 5$ mice) or 28 ET neurons (11−45 neurons, $n = 6$ mice) (Fig. 2c).

We first examined the responses of IT and ET neurons to CS+ and CS− in naïve mice prior to training (Fig. 2d−g). A larger fraction of IT neurons responded to whisker stimuli compared to ET neurons (56.41% vs. 35.85%, $n = 273$ IT neurons vs. 159 ET neurons; $P = 3.7 × 10^{-5}$, $\chi^2 = 17.00$; Fig. 2e, g). While subsets of IT neurons exhibited selectivity for either CS+ or CS−, a notable fraction of neurons responded to both stimuli (Fig. 2e). Smaller but comparable proportions of IT and ET neurons responded to water rewards delivered independently of whisker stimuli (15.26% vs. 18.47%, $n = 262$ IT neurons vs. 157 ET neurons; $P = 0.39$, $\chi^2 = 0.73$; Fig. 2e, g). To assess the reliability of stimulus encoding in each population, we performed a population-decoding analysis using a linear support vector machine (SVM). Decoders trained on IT neuronal responses reliably predicted stimulus identity

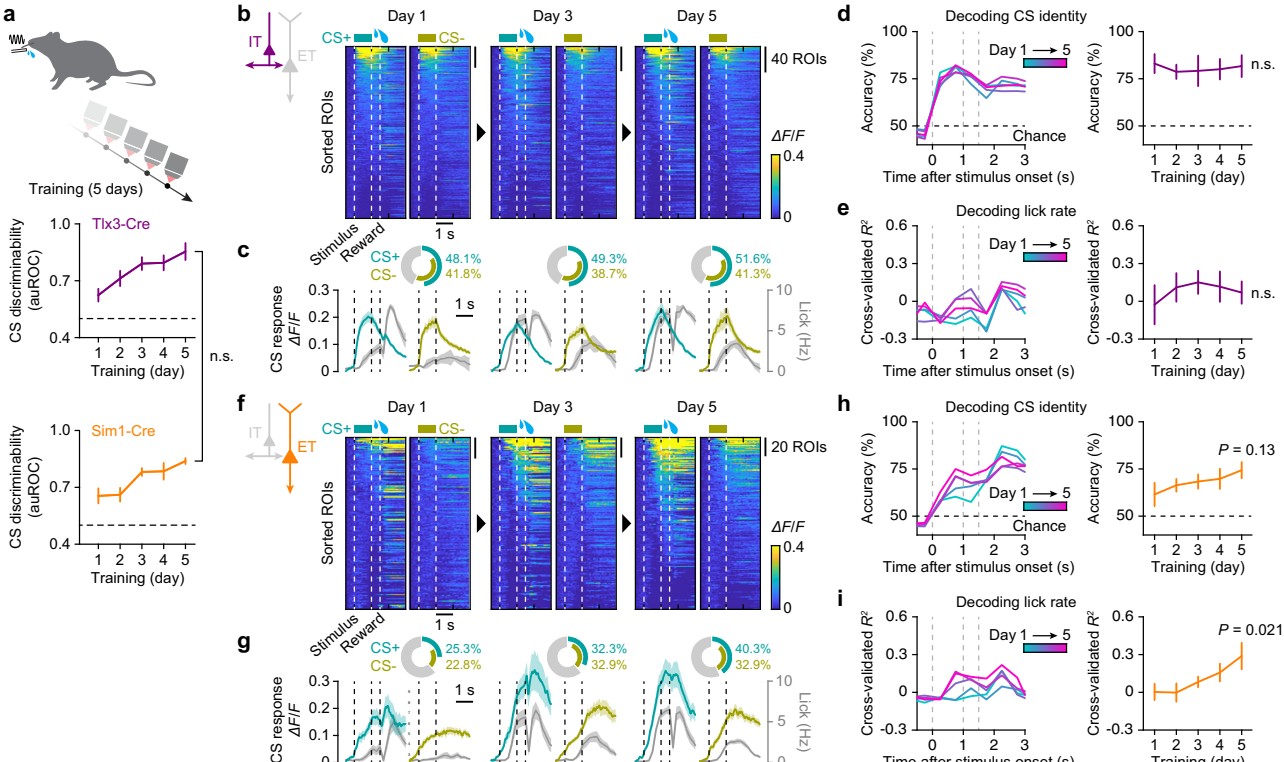

**Fig. 3 | L5 IT neurons maintain robust stimulus encoding, while ET neurons develop reward-expectation responses during learning. a** Top, longitudinal imaging of IT and ET neuronal activity during the course of training on the behavioral task for 5 consecutive days. Bottom, evolution of CS discrimination performance of Tlx3-Cre mice ($n = 5$ mice) and Sim1-Cre mice ($n = 6$ mice). $P = 0.96$, $F = 0.0032$; two-way mixed ANOVA. **b** Heatmaps of trial-averaged responses of IT neurons during learning, sorted based on the mean response amplitudes from the stimulus onset until the reward onset at each day ($n = 5$ mice). **c** Average calcium responses across IT neurons responding to CS+ (blue) or CS− (yellow), overlaid with average lick rates (gray). Inset, pie-charts showing the fractions of IT neurons

responding to CS+ or CS−. **d** Left, SVM decoder performance in classifying the trial types (CS+ or CS−) across time within the trial over training Days 1–5, based on IT neuronal responses ($n = 5$ mice). Right, decoder accuracies for the time window of 0–1.5 s after the stimulus onset ($P = 0.79$, $F = 0.42$; one-way repeated-measure ANOVA). **e** Left, SVM regression performance in decoding lick rates over training Days 1–5, based on IT neuronal responses ($n = 5$ mice). Right, regression performance for the time window of 0–1.5 s after the stimulus onset ($P = 0.85$, $F = 0.33$; one-way repeated-measure ANOVA). **f**–**i** Same as (**b**–**e**) but for ET neurons ($n = 6$ mice). **h** Right, $F = 2.01$. **i** Right, $F = 5.63$. Data were presented as mean ± SEM. Source data are provided as a Source Data file.

on a trial-by-trial basis, whereas decoders trained on ET neuronal responses performed at chance level (Fig. 2h). The finding that some ET neurons show stimulus selectivity while population decoding remains poor indicates that ET population activity is sparse and unreliable at the single-trial level.

Together, these results reveal distinct baseline response profiles of IT and ET neurons in the naïve condition. IT neurons exhibited robust stimulus representations sufficient to reliably encode stimulus identity, whereas ET neurons responded to stimuli but did not distinguish between them. By contrast, reward responses were present in both populations, though in a smaller fraction of neurons.

## IT population maintains robust stimulus coding during learning, while ET population acquires reward-expectation signals

To determine how IT and ET neuronal activity evolve during learning, we imaged the same mice across consecutive daily training sessions. Both Tlx3-Cre and Sim1-Cre transgenic mice learned the task at a comparable rate over 5 days (Fig. 3a). The response profile of IT neurons remained remarkably stable throughout learning (Fig. 3b, c). A large fraction of IT neurons was stimulus-responsive, and their activity was tightly time-locked to stimulus onset. Neither the fraction of stimulus-responsive neurons nor their response amplitudes changed during learning (Fig. S2a, b). As in naïve mice (Fig. 2e), most IT neurons responded to both CS+ and CS−, with a subset of neurons showing stimulus specificity (Fig. 3c and Fig. S2a). We assessed the stimulus

encoding capacity of IT neurons using a population-decoding analysis with a linear SVM classifier. Stimulus identity (CS+ vs. CS−) was decoded throughout the trial using time-binned population activity (bin = 0.5 s) across Days 1–5. Regardless of training day, decoding accuracy consistently rose sharply following stimulus presentation and remained high throughout the trial (Fig. 3d). We further tested whether IT population activity encoded licking behavior by performing SVM regression analysis to predict lick events throughout the trial. The decoder showed weak performance in predicting licking activity, including anticipatory licks during the stimulus and trace intervals, and this weak performance did not improve during learning (Fig. 3e).

In contrast to the stable, stimulus-locked responses of IT neurons, ET neurons underwent striking, learning-dependent changes. As training progressed, their activity during the stimulus and trace intervals ramped towards the expected moment of reward rather than locking to stimulus onset (Fig. 3f, g). Both the proportion of responsive ET neurons and their response amplitudes increased with learning, paralleling the emergence of anticipatory licking (Fig. 3g and Fig. S2c, d). Accordingly, SVM classifiers trained on ET population activity decoded the stimulus identity only weakly on Day 1, but decoding accuracy tended to improve as training advanced (Fig. 3h). This improvement was already evident relative to naïve mice (Fig. 2h), consistent with the animals beginning to acquire the task on Day 1 (Fig. 3a). Importantly, the same population became significantly effective at predicting anticipatory licking during the stimulus and trace intervals,

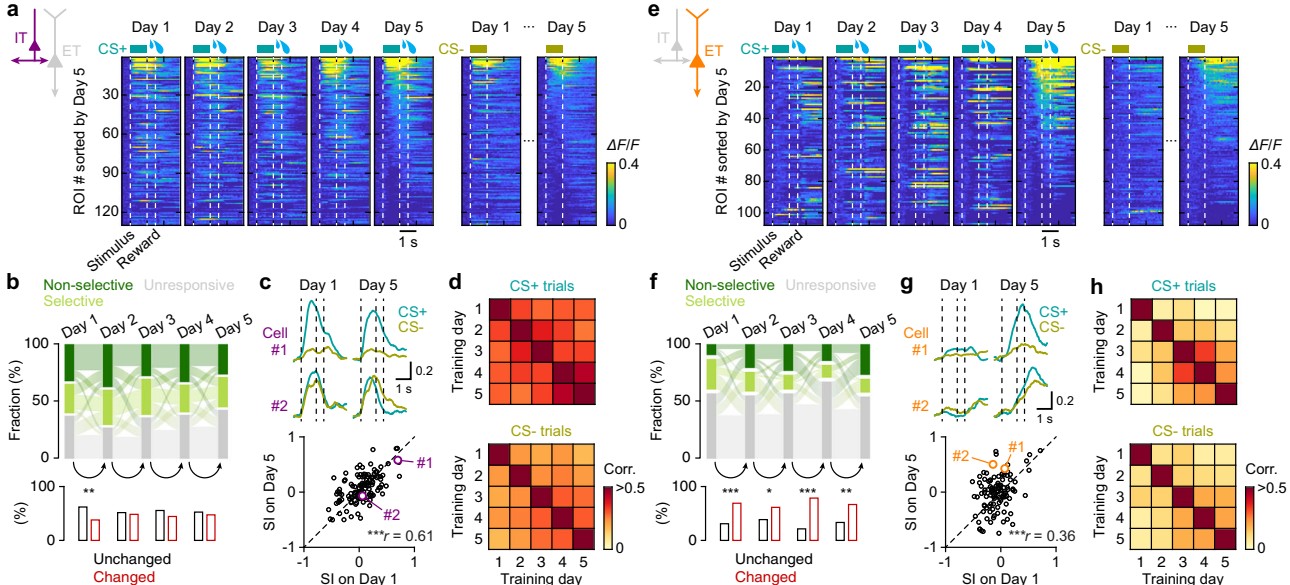

**Fig. 4 | Static IT neuronal encoding and adaptive ET neuronal responses throughout learning. a** Heatmaps of trial-averaged responses of tracked IT neurons during learning, sorted based on the calcium response amplitudes at Day 5 ($n = 130$ neurons from five mice). **b** Top, Sankey diagrams showing day-to-day transitions of tracked IT neuron response types across training (non-selective: active in both CS+ and CS− trials; selective: active in either CS+ or CS− trials; non-responsive: inactive in both). Bottom, bar plots showing the fraction of responsive neurons that maintained (black) or changed (red) their response types between days (Day 1–2: $^{**}P = 0.0025$; $2 \times 2\ \chi^2$ test). **c** Top, example response patterns of two tracked IT neurons on Day 1 and Day 5. Bottom, selectivity index (SI), compared

between Day 1 and Day 5 ($^{***}P = 1.5 \times 10^{-14}$; Pearson's correlation). SI >0 indicates preference for CS+, while SI <0 for CS− (see Materials and methods for details). SI values for the example neurons shown above are highlighted in color. **d** Pairwise Pearson's correlation coefficients of trial-averaged responses between training days for tracked IT neurons during CS+ trials (top) and CS− trials (bottom). Values were averaged across all neurons within each mouse, then across mice. **e**–**h** Same as (**a**–**d**) but for tracked ET neurons ($n = 108$ neurons from six mice). **f** Day 1–2: $^{***}P = 4.8 \times 10^{-4}$; Day 2–3: $^{*}P = 0.033$; Day 3–4: $^{***}P = 1.6 \times 10^{-7}$; $^{**}$Day 4–5: $P = 0.0098$. **g** $^{***}P = 1.4 \times 10^{-4}$. Source data are provided as a Source Data file.

consistent with idea that ET neurons track the emergence of reward anticipation (Fig. 3i).

Alternatively, these responses could simply reflect lick counts or motor execution[30], though several observations argue against this interpretation. First, decoders trained solely on CS+ trials failed to predict the number of anticipatory licks (Fig. S3a), indicating that ET neurons do not represent individual lick events. Second, the onset of ET neuronal activity was not time-locked to the first lick across trials (Fig. S3b, c). Third, neurons active during anticipatory licking were not necessarily the same as those recruited during consummatory licking after reward delivery (Fig. S3d). Accordingly, decoders trained on anticipatory licks performed poorly when applied to consummatory licks, and vice versa (Fig. S3a). Together, these results strengthen the conclusion that ET activity is not a simple readout of licking but instead reflects a reward-expectation signal. In line with this, we interpret the improved SVM decoding of CS identity during learning not as a consequence of enhanced sensory responsiveness of ET neurons, which is weak in naïve mice (Fig. 2f–h), but rather as a readout of emerging behavioral performance through reward-expectation signals.

Collectively, these findings suggest that IT neurons provide stable sensory coding that changes little with learning, whereas ET neurons acquire a learning-dependent ramping signal consistent with reward expectation, not merely lick execution.

## IT neurons preserve stable response patterns across days, while ET neurons undergo dynamic reorganization

To assess how single-neuronal activity changes with learning, we tracked neurons that could be reliably identified across all five daily imaging sessions. Across training, individual IT neurons displayed remarkably stable response patterns (Fig. 4a and Fig. S4a). From day-to-day, neurons that were already responsive to stimuli mostly

remained responsive (Fig. 4b). Consistently, their stimulus selectivity on Day 1 closely matched that on Day 5 (Fig. 4c). Pearson correlations of session-averaged activity traces were consistently high across all pair of days and for both CS+ and CS− trials, confirming the stability of IT responses (Fig. 4d). Most IT neurons therefore maintained stable stimulus responses, although a small subset with baseline reward responses prior to training selectively increased their responses to CS+ during learning (Fig. S5).

In contrast, ET neurons exhibited marked day-to-day variability: response patterns shifted, new neurons were recruited, and previously responsive neurons often fell silent (Fig. 4e–g and Fig. S4b). These dynamics resulted in lower inter-day correlations, which nevertheless increased gradually across sessions, indicating that ET activity patterns became progressively more consistent with learning (Fig. 4h).

Together, these results suggest that individual IT neurons maintain stable, stimulus-locked activity patterns that are largely unaffected by learning, whereas individual ET neurons undergo dynamic reorganization that progressively stabilizes with learning.

## Distinct contributions of IT and ET neurons to learning

The contrasting activity changes of IT and ET neurons during learning suggests a division of labor between these L5 neuronal subtypes in learning. To test this idea, we selectively silenced each subtype in the C2 barrel column. We injected a Cre-dependent viral vector encoding hM4Di, an inhibitory designer receptor exclusively activated by designer drug (DREADD), into Tlx3-Cre or Sim1-Cre mice (Fig. 5a, b)[31]. Prior to each training session, a DREADD ligand, clozapine-*N*-oxide (CNO), was systemically injected into the mice, thus selectively inhibiting the targeted neuron types. Silencing either IT or ET neurons markedly impaired acquisition of the task compared to control mice with intact S1 (Fig. 5c–e). This impairment was not due to off-target effects of CNO (Fig. S6a).

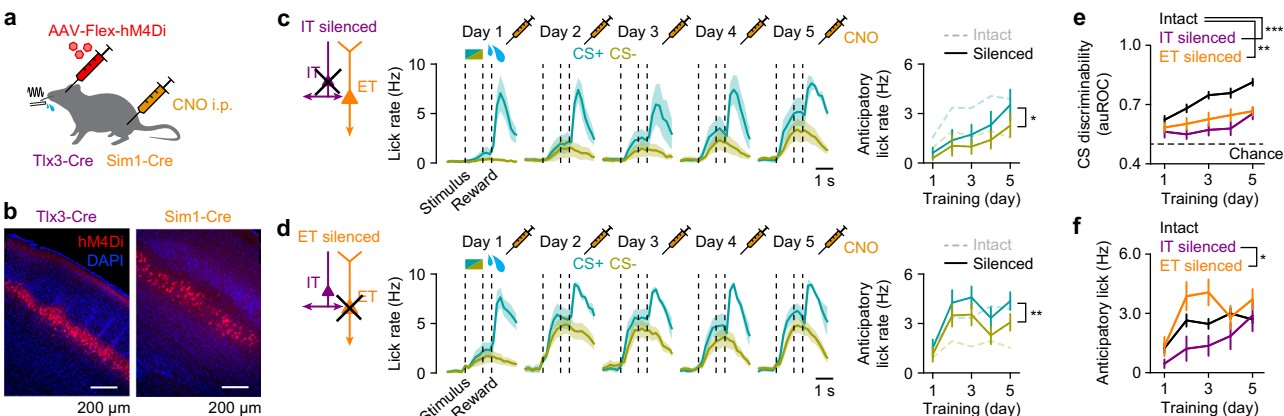

**Fig. 5 | Silencing IT or ET neurons disrupts learning, but with different effects.**
**a** Targeted chemogenetic silencing of IT or ET neurons in S1 in Tlx3-Cre or Sim1-Cre mice, respectively. **b** Example image of a coronal section of S1 where IT (left) and ET neurons (right) expressed hM4Di-mcherry. **c** Left, average lick rates for CS+ (blue) and CS− (yellow) trials across mice with silenced IT neurons for Days 1–5 ($n = 6$ mice). Dashed lines represent CS onset, offset and reward onset. Right, evolution of mean anticipatory lick rates during learning (*$P = 0.030$, $F = 8.94$; two-way repeated-measure ANOVA). Note that the difference in anticipatory lick rates between CS+ and CS− was markedly reduced compared to intact S1 (Fig. 1c).

**d** Same as (**c**) but for mice with silenced ET neurons ($n = 6$ mice; **$P = 0.0016$, $F = 38.74$). **e** Behavioral performance of mice with intact S1 ($n = 20$ mice), silenced IT neurons ($n = 6$ mice) and silenced ET neurons ($n = 6$ mice; $P = 1.9 \times 10^{-5}$, $F = 16.15$; two-way mixed ANOVA with post hoc Tukey–Kramer test). **$p < 0.01$, ***$p < 0.001$. **f** Mean anticipatory lick rates of mice with intact S1 ($n = 20$ mice), silenced IT neurons ($n = 6$ mice) and silenced ET neurons ($n = 6$ mice; $P = 0.034$, $F = 3.80$; two-way mixed ANOVA with post hoc Tukey–Kramer test). *$p < 0.05$. Data were presented as mean ± SEM. Source data are provided as a Source Data file.

Closer inspection of anticipatory licking revealed divergent effects of chronic inactivation during training (Fig. 5f). Compared to control mice (Fig. 1c), silencing IT neurons led to an overall reduction in anticipatory lick rates for both CS+ and CS− (Fig. 5c), with a more pronounced decrease for CS+ (Fig. S6b), ultimately abolishing discrimination between the stimuli. In contrast, silencing ET neurons produced a marked increase in anticipatory licking to both CS+ and CS−, most prominently during the first two training days, and this elevation persisted across sessions (Fig. 5d). The effect was particularly strong for CS− (Fig. S6c), resulting in poorer discrimination relative to controls.

Chemogenetic inhibition of either neuron type in expert mice had no effect on performance (Fig. S6d–g), consistent with the lack of impact seen with global S1 inactivation by muscimol (Fig. 1i). Notably, ET neuronal silencing did not alter the learned pattern of licking (i.e., increased anticipatory licking to CS+ and reduced licking to CS−; Fig. S6f). Thus, the indiscriminate licking observed during training is not a general effect of ET silencing, but reflects a specific impairment of the learning mechanism needed to differentiate between conditioned stimuli.

Together, these results suggest distinct functional roles for IT and ET neurons in learning, where IT neurons are essential for forming the basic stimulus-reward association, whereas ET neurons are required to refine that association into accurate CS discrimination.

### A reinforcement learning model reflecting L5 subnetworks captures learning dynamics

The differences in response properties in IT and ET neurons and their impact on learning imply distinct computational roles in learning stimulus-reward associations. To explore this further, we employed mathematical modeling to decipher their specific contributions. The Rescorla–Wagner (RW) model is a simple yet widely used reinforcement learning framework for understanding classical Pavlovian conditioning[32]. The basic principle of the RW model is to update the value function (i.e., association strength) for each conditioned stimulus by calculating the reward prediction error (RPE) through learning. We implemented an RW-type model within a neural network framework, in which the neural network was trained to predict the value of each stimulus type (CS+ vs. CS−) using an RW update rule, here

referred to as "value-encoding network" (Fig. 6a). Next, we extended this model to incorporate three key features observed in our experimental results. First, IT neurons, which exhibited stable representations of stimuli before learning (Figs. 2, 3), were modeled as a neural network that was pretrained in an unsupervised learning task to reconstruct (see Methods) and remained unchanged during stimulus-reward training. After unsupervised training, the IT neural network developed representations that help differentiate between stimuli, thus helping the value-encoding network in predicting the reward values (Fig. 6a, left). Second, based on our experimental results showing that reward-predicting responses to stimuli evolve in ET neurons over learning (Fig. 3), we modeled ET neurons as relaying reward-predicting signals generated by the value-encoding network to an RPE calculation circuit (Fig. 6a, right). These signals are then compared with actual rewards, generating RPEs that update the value-encoding network during reward-based training. Lastly, since expert mice were able to perform the task without S1 (Fig. 1i and Fig. S1b, c), the learned associations (i.e., stimulus values) must therefore be stored in brain regions outside of S1. These regions should therefore also receive sensory input and are capable of executing learned responses independently of S1 during expert-level performance. To model this, the value-encoding network includes two input channels: "IT" and "non-S1" (Fig. 6a, left). Both channels independently transmit sensory information to predict stimulus values. This component of the model predicts that the IT channel is responsible for conveying richer stimulus representations coming from the IT network, while the non-S1 channel only conveys coarse stimulus representations (see Methods).

The model was trained to associate two distinct classes of artificial stimuli, representing CS+ and CS−, with the correct reward outcome. The predicted anticipatory lick rate was derived by linearly scaling the association strength, or value, of each stimulus. Comparison with experimental data showed that the model accurately reproduced the learning dynamics across conditions. Specifically, in the control condition, the model closely tracks the tendency in the experimental data for the initial rise in lick rates for both stimuli, followed by a divergence where CS+ licks continued to increase, and CS− licks declined (Fig. 6b). Analysis of the IT neural network in the model revealed that this early general increase could be attributed to the proximity of CS+ and CS−

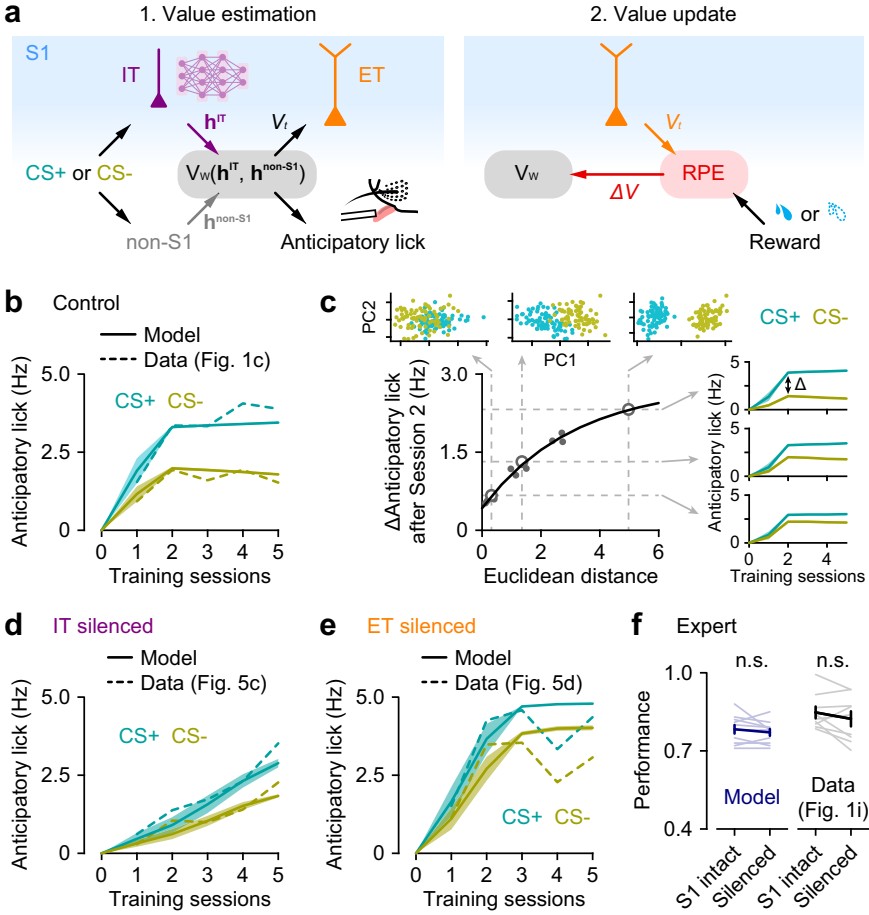

**Fig. 6 | A reinforcement learning model of L5 subnetworks reproduces learning dynamics across conditions. a** Schematics of the Rescorla−Wagner (RW)-type model, which implements two steps: value estimation and value update. Left, a pretrained IT network (purple) processes the stimulus, providing a stimulus representation ($\mathbf{h}^{IT}$), with which the value-encoding network ($V_W$) estimates the value of the stimulus ($V_t$). Additionally, the value-encoding network also receives a stimulus representation, $\mathbf{h}^{non-IT}$, independently of S1. The predicted value is also used to model the anticipatory lick rate. Right, ET neurons convey the predicted value, which is then compared with the actual reward to calculate the reward prediction error (RPE). Finally, the RPE is sent back to the value-encoding network to update the prediction function. **b** Association strengths (solid lines) from the model for CS+ and CS− stimuli through training in the control condition ($n = 10$ simulations), plotted together with anticipatory lick rates (dashed lines) from the experimental data. **c** Relationship between learning dynamics and overlap

between CS+ and CS− stimuli in the IT network representational space (gray dots), with an exponential fit (black line). Learning dynamics is represented by the difference in association strengths (i.e., anticipatory licks) between distinct CS+ and CS− pairs after Session 2. Representational overlap is measured as the Euclidean distance between CS+ and CS− in the IT network space. Inset, the first two principal components (PCs) of IT representations for three example CS+ and CS− pairs, marked as open circles in the main panel, and their corresponding association strengths across training. **d** Same as (**b**) but for the condition where IT neurons were silenced ($n = 10$ simulations). **e** Same as (**b**) but for the condition where ET neurons were silenced ($n = 10$ simulations). **f** Discrimination performance in experts before and after silencing of S1, from the model (blue; $n = 10$ simulations; $P = 0.35$, two-sided paired $t$-test) and the experimental data (black; $n = 9$ mice; $P = 0.23$, two-sided paired $t$-test). Data were presented as mean ± SEM. Source data are provided as a Source Data file.

stimuli in the encoding space (Fig. 6c), suggesting that overlapping stimulus representations in IT neurons contribute to the general increase observed in the experiments (Fig. 5c). The model also captured the distinct learning dynamics observed when IT or ET neurons were blocked. Because in the model IT neurons are responsible for rich sensory encoding, blocking them resulted in a general slower learning for both stimuli (Fig. 6d). On the other hand, because ET neurons mediate the transmission of value for RPE computations blocking highlights the fast sensory learning mediated by IT neurons, but inability to correctly assign values to specific stimuli (Fig. 6e). Therefore, our model shows that comparing ET neuronal output against the actual reward outcome to generate a RPE is consistent with our data (Fig. 5d). Furthermore, it indicates the potential involvement of an RPE calculation circuit downstream of ET neurons. Finally, as expected, our model exhibits learning transfer, so that as learning progresses, the non-S1 input channel in the value-encoding network increasingly builds stimulus-reward associations, leading to a reduced reliance on the IT input channel. This aligns with our experimental observation

that, at the expert level, neither IT nor ET neurons in S1 are required for task execution (Fig. 6f).

In conclusion, the model effectively captured the distinct contributions of IT and ET neurons to stimulus-reward learning and accurately mirrored experimental findings. These results highlight the complementary computational roles of IT and ET neurons, which fit within a classical associative learning model, where IT neurons provide representations of sensory stimuli to estimate stimulus value, and ET neurons relay this value to the prediction error regions for updating the value function.

## Discussion
Recent studies have begun to reveal distinct roles for L5 IT and ET neurons in cortical sensory areas in decision-making, perception, and cognitive processes[4,5,16,17,21]. Our study extends the understanding of these roles by revealing the unique contributions of each L5 subtype to stimulus-reward associative learning. Specifically, we found that IT neurons have stable and robust responses to the stimuli from the

beginning which are maintained throughout the learning process, regardless of the associated stimulus value. This indicates their critical role in reliably encoding sensory information. In contrast, ET neurons showed dynamic changes, with their activity patterns evolving as learning progressed. Their responses to the stimuli became increasingly pronounced in parallel with the development of anticipatory licking, suggesting that ET neurons signal the behavioral relevance of stimuli and the expectation of reward.

Our chemogenetic inactivation experiments reinforced the observed distinctions. Silencing IT neurons markedly impaired the development of anticipatory licking to both CS+ and CS−, highlighting their essential role in forming stimulus-reward associations. In contrast, silencing ET neurons did not disrupt the overall increase in anticipatory licking but selectively prevented the suppression of licking to CS−, underscoring their role in refining and differentiating learned responses.

Moreover, these findings align seamlessly with the classical reinforcement learning framework. Our modeling results suggest that IT neurons encode sensory stimuli based on a pre-acquired world model, facilitating value estimation that is crucial for learning. In contrast, ET neurons are engaged in transmitting this estimated value to compute prediction errors, a process fundamental to refining associations. Thus, L5 IT and ET neuronal subpopulations in sensory cortex appear to contribute in distinct yet complementary ways to stimulus-reward associative learning. We note, however, that our modeling framework is necessarily simplified and does not exclude the possibility that alternative models could also capture the data. Rather than serving as a definitive mechanistic account, the model provides a conceptual bridge between our empirical findings and reinforcement learning theory. In this light, the proposed division of labor between IT and ET neurons should be viewed as a testable hypothesis that can be further refined by future experiments directly probing their respective roles in value estimation and prediction error calculation across different learning contexts. For example, a key prediction of our model is that directly manipulating CS representations encoded by IT neurons would specifically alter the reward-predicting signal in ET neurons. As we have demonstrated the ability to selectively modulate L5 IT neuronal activity in the present study, this provides a feasible path for future research to directly test this prediction.

## Involvement of S1 in stimulus-reward learning

Pharmacological inactivation of S1 severely impaired learning (Fig. 1h), while it did not disrupt the performance in expert mice (Fig. 1i and Fig. S1b, c). This observation aligns with previous studies, indicating that sensory cortex is critical during learning but not necessarily required for the performance of well-established, habitual responses, particularly for simple sensory tasks[5,33,34]. Importantly, our results imply that learned stimulus-reward associations are maintained outside of S1.

The striatum, a central region involved in reward-based learning[35], emerges as a likely candidate for the retention of learned associations, as lesions in this area have been shown to abolish conditioned responses even in fully trained animals[33]. Both IT and ET neurons in the sensory cortex project to the striatum. Previous studies have shown that stimulus-reward learning induces synaptic strengthening in cortico-striatal projections[3] and blocking these pathways disrupts learning[5], but until this study the specific roles of IT and ET neurons in this process had not been investigated. Given the results of this study, including the fact that IT and ET neuronal output are quite dissimilar, we hypothesize that they each exert different effects on striatal activity during learning, particularly in the context of value estimation and updating of learned associations.

## Contribution of IT neuronal activity to stimulus-reward learning

What are the implications of the clearly distinct response patterns and changes seen in IT vs. ET neurons? A large proportion of the IT neurons responded to the stimuli and were able to distinguish the stimulus types with high accuracy, and their response patterns remained largely unchanged during learning. Extrapolating from these observations, we hypothesize that the IT population forms a pretrained network that encodes various features with high precision. Nevertheless, it was also notable that the majority of stimulus-responding IT neurons responded to both CS+ and CS−. Our model suggests that this overlap in representations contribute to the generalized stimulus-reward associations that mice initially acquire during learning (Fig. 6c). The representational overlap between similar stimuli could promote generalized learning. In some cases of learning, it may reflect a biological strategy to quickly establish a generalized association between stimulus and outcome per se, and then to take a longer time to establish stimulus-specific differences that more closely predict outcome.

## Contribution of ET neuronal activity to stimulus-reward learning

In contrast to IT neurons, which maintained stable stimulus-locked responses, ET neurons displayed dynamic learning-dependent changes. Their activity increased during the stimulus and trace intervals in parallel with the emergence of anticipatory licking, suggesting it encodes behaviorally-expressing reward expectation. Several observations argue against the alternative hypothesis that ET neuronal activity simply reflects a motor command (Fig. S2): ET responses did not reliably track individual lick events, were not time-locked to lick onset, and neurons engaged during anticipatory licking were distinct from those active during consummatory licking. Moreover, silencing ET neurons during learning, but not after, impaired task performance, further supporting their role in acquiring reward-expectation signals rather than simply mirroring licking behavior. This interpretation is consistent with a previous report showing that dendrites of L5 neurons do not respond to spontaneous licks unrelated to rewards[36]. Together, these findings indicate that ET neurons undergo a dynamic reorganization during learning that enables them to convey reward-expectation signals.

Reward-expectation signals are fundamental to the process of reinforcement learning, particularly in refining stimulus-specific responses in associative learning. According to classical theoretical models of reinforcement learning[1,37,38], prediction errors are computed by comparing reward-predicting signals with the actual reward availability, reinforcing responses to CS+ and suppressing responses to CS−. The idea that the output of ET neurons is used to calculate the reward prediction error necessary for refining the learned responses is consistent with the finding that the mice with silenced ET neurons showed impaired learning to discriminate between CS+ and CS− (Fig. 6a, e). However, since the initial association was unaffected by the inactivation of ET neurons, it suggests that their output is not critical for generating positive prediction errors, i.e., positive reinforcement. Instead, they are more likely to be involved in the generation of negative prediction errors, thereby suppressing incorrect responses, such as licking to CS−. Identifying the downstream circuits that compute prediction errors using the output of ET neurons would be an important area for future research. One potential target is the subcortical dopaminergic circuits[39]. Recent reports indicate that transient drops in subcortical dopamine signals are responsible for negative prediction errors and contribute to the attenuation of CS− responses in discrimination learning[40]. The reward-expectation signals transmitted by ET neurons may contribute to the generation of these dopamine dips.

Although ET responses gradually stabilized across training, individual neurons showed substantial day-to-day variability in their response profiles. This variability suggests that downstream circuits likely rely on the collective activity of the ET population rather than on specific neurons. In this view, even if the subset of responsive ET neurons shifts across days, the population-level signal representing

reward expectation remains stable. Our model is consistent with this principle. Determining how downstream targets integrate this population signal will be an important direction for future work.

How reward-expectation signals are generated in ET neurons remains to be investigated in future research. The apical dendrites of L5 neurons, located in cortical layer 1 (L1), are hypothesized to function as critical sites for cortical association mechanisms[41–45]. L1 receives non-sensory top-down inputs from higher cortical areas which may contribute to the formation of reward-expectation signals. The orbitofrontal cortex (OFC) is an important structure in value encoding during stimulus-reward associations[46–48]. A recent study by Liu et al. demonstrated that top-down inputs from the OFC are involved in the formation of reward-expectation activity within the sensory cortex by modulating the activity of dendrite-targeting inhibitory neurons[49]. Inhibition of top-down inputs from the OFC during learning prevented mice from suppressing incorrect lick responses to reward-irrelevant stimuli, paralleling our results from ET neuronal suppression. Thus, the pathway from the OFC to L5 ET neurons in the sensory cortex may convey reward-expectation signals to downstream areas, facilitating the generation of negative prediction errors, essential for refining behavioral responses.

In conclusion, we show that distinct subtypes of L5 neurons in sensory cortex are involved in learning, albeit in different aspects of learning. Overall, IT neurons showed stable responses conveying stimulus information necessary for forming general associations between stimuli and reward. In contrast, ET neuronal responses evolved over learning, conveying reward-expectation signals, which were necessary for learning to discriminate between stimuli. Their unique activity patterns highlight parallel cortical computations during learning and demonstrate distinct yet complementary contributions of IT and ET neurons to associative learning[12].

## Methods
### Animals
Adult C57BL/6J wild-type mice (male and female) or Sim1-Cre (KJ18) (MMRRC, no. 031742-UCD) and Tlx3-Cre (PL56) (MMRRC, no. 041158-UCD) transgenic mice (>P60) were used. Mice were housed in groups of 2–4 mice per cage in a 12:12 reversed day-night cycle. All experiments were conducted following the guideline given by Landesamt für Gesundheit und Soziales Berlin (LAGeSo) and were approved by this authority (protocol number: G0278/16 and G0161/21).

### Surgical procedures
During the surgery, mice were anesthetized with isoflurane (1.5–2.0% in O$_2$) or ketamine (100 mg/kg)/xylazine (10 mg/kg) and kept on a thermal blanket. The skin was removed and the skull was carefully cleaned and scraped with a surgical scalpel. A light-weight head-post was fixed on the skull over the right hemisphere with light-curing adhesives and dental cement. To locate the C2 barrel column for viral injection and pharmacology, we performed intrinsic signal imaging. Mice were lightly anaesthetized with isoflurane (0.8–1.5% in O$_2$) during imaging. The C2 whisker was inserted into a small metal tube attached to an electromagnet-based actuator and deflected at 10 Hz. Imaging was performed through the intact skull, which was covered with lukewarm phosphate buffer solution (PBS, 0.1 M) and a small cover-glass placed on top. Under red LED illumination, changes in reflectance were recorded with and without whisker stimulation, and the difference image was used to identify the C2 barrel column using custom-written LabView software. Both the baseline (no stimulation) and the stimulus period were set for 5.5 s.

For chemogenetic inhibition, AAV2/1-hSyn-DIO-hM4D(Gi)-mCherry (viral titer: 2.3 × 10$^{13}$; Addgene, Product #44362) was injected through a glass pipette (tip diameter, 5–10 μm) into the contralateral (left) S1 (100 nL at 700-μm depth under the pia). The experiments started 3 weeks after viral injection. As a control viral vector, AAV9-EF1a-DIO-mCherry-WPRE3 (viral titer: 1.91 × 10$^{12}$; The Viral Core Facility of the Charité - Universitätsmedizin Berlin) was used.

For in vivo two-photon calcium imaging, AAV2/1-syn-FLEX-jGCaMP8m-WPRE (viral titer: 5.0 × 10$^{12}$; Addgene, Product #162378) was injected through a glass pipette (tip diameter, 5–10 μm) into the contralateral (left) C2 barrel column (100 nl at 700-μm depth under the pia). A 3-mm diameter craniotomy was made over the injected area, and the skull and dura were carefully removed. The craniotomy was sealed with a triple-layered glass coverslip (3 mm diameter for the two bottom layers and 4 mm diameter for the top layer) with cyanoacrylate glue and dental cement. Experiments started 3 weeks after viral injection.

### Behavior
Mice were kept on a reversed light/dark cycle. Habituation of the mice to head restraint began a week after the head-post surgery. Head-restrained time on the first day was 5 min and then gradually increased each day until the mice sat calmly for an hour. Mice were water-restricted during habituation and subsequent periods of behavioral training.

Behavioral events (e.g., licking, whisker deflection, reward delivery) were monitored and controlled by a custom-written program running on a microcontroller board (Arduino). The C2 whisker was deflected by displacing a light metal tube (~3 mg) slid over the whisker using a magnetic coil placed underneath the animal. A sinusoidal current applied to the coil generated a local magnetic force that produced ~10–20° whisker deflections, about an order of magnitude above threshold stimuli[50], ensuring the stimuli were well above the animals' perceptual threshold. Mice were exposed to one of two different frequencies of whisker deflection, 10 and 5 Hz, with a duration of 1 s. The 10 Hz whisker deflection (CS+) was paired with a water reward (~5 μl), delivered through a lick port 0.5 s after the stimulus offset, while the 5 Hz whisker deflection (CS−) was not followed by any reward. Licking was detected by a piezo-electric device attached to the licking spout. The mice were allowed to lick at any time, and no punishments were given for any premature licking event. During CS+ trials, we frequently observed a brief pause in licking upon water delivery. This interruption may arise from the behavioral shift between anticipatory and consummatory licking, or from a technical limitation: the animal could lick the dispensed droplet without touching the spout, causing the piezo sensor to miss those licks. Each mouse received 100 trials of each stimulus type (200 trials in total, each trial having an inter-trial interval of 6–8 s), each session repeated for 5 days. The probability of receiving CS+ or CS− was 50% during each session.

For in vivo calcium imaging, there was a pre-training imaging session 1 day prior to training. This session was further separated into two blocks, one where mice were exposed to the stimuli only (without reward) and one where mice were given reward-alone (without stimuli).

### Behavioral performance analysis
To assess the performance of the task, we used the anticipatory licking occurring from the stimulus onset to the time of the reward (1.5 s) to calculate the area under a receiver operating curve (auROC). The auROC score demonstrates how well the lick patterns in CS+ and CS− trials overlap across one session. The score ranges from 0 to 1 where auROC >0.5 indicates higher amount of anticipatory licks in the CS+ trials and auROC <0.5 indicates higher amount of anticipatory licks in the CS− trials. The initial 10% of trials were discarded from the analysis to avoid impulsive licking behavior by the mice at the beginning of each session. To compare behavioral performances across cohorts, we performed a two-way mixed ANOVA where we focused on the comparison between mouse groups.

### In vivo pharmacology

Before every training session, the mice were lightly anesthetized with isoflurane (1.5–2.0% in $O_2$). A very small craniotomy was made above the C2 barrel column, and muscimol (5 mM, Tocris) was injected at two depths (at depths of 700 and 350 μm, 100 nl each). The craniotomy was sealed with a silicone sealant (Kwik-Cast, World Precision Instruments). The mice were put back in their homecage to recover for 10 min before commencing the behavior. After the last training session of the pharmacology experiment, fluorescent muscimol (BODIPY TMR-X conjugate, Thermo Fisher Scientific) was injected into the same injection site (at depths of 700 and 350 μm, 100 nl each), and mice were perfused 30 min after the injection.

### In vivo chemogenetics

The experiments started 3 weeks after viral injection of hM4Di. Before each session, mice were injected with CNO (5 mg/kg intraperitoneally, Tocris) under brief anesthesia with isoflurane and kept for 30 min in a homecage prior to behavioral testing.

### Histology

Mice were anesthetized using isoflurane (1.5–2% in $O_2$) and euthanized by an intraperitoneal injection of urethane (1.5 g/kg). Mice were perfused transcardially with 0.1 M PBS, followed by 4% paraformaldehyde (PFA) in PBS. After perfusion, brains were removed from the skull and postfixed in PFA overnight. The next day, brains were washed in PBS, transferred into a 30% sucrose solution in PBS, and left for 24–48 h for cryoprotection. For cryosectioning, brains were embedded in optimal cutting temperature compound. Coronal brain sections (70 μm) were washed twice in PBS for a minimum of 10 min each at room temperature before staining the nuclei were stained using DAPI (NucBlu Fixed Cell ReadyProbe Reagent, Thermo Fisher). After washing, sections were mounted on slides and coverslipped with Fluoromount-G mounting medium. Images were obtained using a fluorescent microscope (DMI 4000B, Leica Microsystems).

### Two-photon calcium imaging

Imaging from behaving mice was performed with a resonant-scanning two-photon microscope (Thorlabs) equipped with GaAsP photomultiplier tubes (Hamamatsu). jGCaMP8m was excited at 940 nm with a Ti:Sapphire laser (Mai Tai eHP DeepSee, Spectra-Physics) and imaged through a 16×, 0.8 NA water-immersion objective (Nikon). Full-frame images (512 × 512 pixels; pixel size, 0.35 × 0.35 μm²) were acquired from the apical trunks of ET or IT neurons expressing jGCaMP8m at a depth from the pia of ~200 μm at 30 Hz using ScanImage 4.1 software (Vidrio Technologies).

### Imaging data analysis

Motion correction of raw files and regions of interest (ROI) selection was performed using Suite2p[51]. Fluorescence change ($\Delta F/F_0$) was calculated for each trial where $F_0$ was the average fluorescence during the trial baseline, i.e., during 1 s prior to stimulus onset. ROIs were considered stimulus-responsive if the time-averaged response during 1.5 s after stimulus onset significantly increased from the baseline activity during 1 s prior to stimulus onset (paired two-sided Wilcoxon signed rank test, alpha = 0.01), and if their normalized $\Delta F/F_0$ response exceeded 0.05. For reward-alone trials, we compared the baseline activity to the mean activity during 1.5 s after reward delivery. To isolate pure stimulus responses and exclude any potential confound of licking, we excluded trials during the stimulus-only blocks in which the mice licked (IT neuronal recording: 0.17 ± 0.41 out of 25 trials; ET neuronal recording: 0.83 ± 1.17 out of 25 trials).

For visualization in traces and heatmaps, calcium traces were smoothed using a moving average of three frames for trial-averaged responses and seven frames for single trials.

For decoding analysis, we tested whether the conditioned stimulus (CS+ or CS−) on each trial could be classified from population neuronal activity using a linear support vector machine (SVM). Trials were binned in 500-ms windows, and decoding performance was evaluated separately for each bin. Decoders were trained and tested using five-fold cross-validation, with the random resampling of training and test sets repeated 100 times per bin. Performance was averaged across iterations. In addition, we assessed whether binned population activity could predict trial-by-trial lick counts using SVM regression. Decoder performance was quantified with five-fold cross-validated $R^2$ values, which range from infinitely negative (indicating poor decoding) to 1 (perfect decoding).

To quantify single-neuron selectivity for stimulus types (CS+ vs. CS−), we defined a selectivity index (SI):

$$SI = (AUC - 0.5) \times 2$$

where AUC is the area under ROC curve computed from calcium responses of individual neurons between CS+ and CS− trials. SI values range from −1 to 1, with negative values indicating a preference for CS + , positive values indicating a preference for CS−, and 0 indicating no preference.

The stability of individual neuronal response was measured by performing a Pearson's correlation coefficient ($r$) for each neuron across five training days. To assess both the stimulus response stability and reward response stability, trial average responses in the time window of 3 s from stimulus onset.

For hierarchical clustering (Fig. S3d), calcium traces were $z$-scored per neuron, and pairwise correlation distances (1 − Pearson's r) were computed, and agglomerative hierarchical clustering was performed using Ward's linkage.

### Computational model

We designed a Rescorla–Wagner-type model[32], which learned the value (association strength) of each stimulus (CS+ vs. CS−) over multiple trials.

We modeled the encoding of different stimuli values, i.e., association strengths, using a simple feedforward "value-encoding neural network" denoted as $V_w$ with synaptic weights, **w**. This network consisted of a single linear hidden layer with 116 units where the network inputs were combined into a unified representation, which was then fed to an output layer with sigmoid activation. The value-encoding network, $V_w$, consisted of two distinct input channels: an S1 IT input channel, providing the IT neuronal representation of the current stimulus (i.e., CS+ or CS− stimuli) and a non-S1 input channel, providing a "raw" representation of the current stimulus independent of S1. At each trial $t$, the value-encoding network prediction for the current stimulus, $\mathbf{x_t}$, is defined as,

$$V_t = V_w(\mathbf{h_t}^{IT}, \mathbf{h_t}^{non-S1})$$

where $\mathbf{h_t}^{IT}$ denotes the IT representation of the current stimulus, $\mathbf{x_t}$, while $\mathbf{h_t}^{non-S1}$ denotes the non-S1 representation of the same stimulus, $\mathbf{x_t}$. The IT representation, $\mathbf{h_t}^{IT}$, was obtained from a separate neural network, the IT network, which was pretrained with unsupervised learning (see below). Conversely, we assumed the non-S1 representation, $\mathbf{h_t}^{non-S1}$, to be the "raw" representation of the stimulus, $\mathbf{x_t}$ (i.e., $\mathbf{h}^{non-S1}_t = \mathbf{x_t}$). Note, the same non-S1 representation $\mathbf{h_t}^{non-S1}$, was also used by a separate value-encoding sub-network to support S1-independent expert performance, as described below.

The value-encoding network adapted its synaptic weights, **w**, to predict the correct values, $V$, for CS+ and CS− stimuli based on the experienced reward outcomes. Following RW models, the value-encoding network synaptic weights were updated at each trial $t$ as

follows,

$$[\Delta \mathbf{w}]_t = \alpha \delta_t \nabla_w V_w$$

where $\delta_t$ denotes the reward prediction error (RPE) at trial $t$,

$$\delta_t = R(\mathbf{x_t}) - V_w(\mathbf{h_t}^{IT}, \mathbf{h_t}^{non-S1})$$

Here, $R$ denotes the reward for the current stimulus, $\mathbf{x_t}$. Note that we assumed the reward to be 1 for CS+ stimuli, while to be 0 for CS−stimuli, i.e., $R(\mathbf{x_t} = CS+) = 1$ and $R(\mathbf{x_t} = CS-) = 0$.

In summary, at each trial $t$, the value-encoding network, $V_w$, took the IT stimulus representation, $\mathbf{h_t}^{IT}$, as well as the non-S1 stimulus representation, $\mathbf{h_t}^{non-S1}$, as inputs and was trained to predict the correct value, $V_t$, for the current stimulus, $\mathbf{x_t}$. Finally, the value-encoding network included a separate sub-network, which could predict the value of each stimulus exclusively based on the non-S1 representation, $\mathbf{h_t}^{non-S1}$. During learning, this network was trained based on the S1 IT-dependent value predictions, $V_w(\mathbf{h_t}^{IT}, \mathbf{h_t}^{non-S1})$, using a simple mean squared loss. In this way, at expert performance, the model could solve the task without requiring S1 IT stimulus representations.

**IT network pretraining.** The IT neural network consisted of a feed-forward auto-encoder architecture with two encoder layers (with 50 units each), a bottleneck layer (with 15 units) and two decoder layers (with 50 units each). This IT network was pretrained via unsupervised learning, using a standard L2 loss between the original and the network-reconstructed stimuli[52]. The stimuli were generated by sampling from eight distinct 20-dimensional Gaussian distributions with distinct random means and equal isotropic covariances. Each Gaussian distribution represented a different stimulus type, such as a specific whisker stimulation frequency, with each sample representing a noisy representation of the corresponding stimulus type. Note, we used Gaussian distributions to represent different stimulus types rather than fixed values to account for noise in sensory processing. Two of the eight distinct types of stimuli were randomly selected to be used as CS+ and CS− stimuli in the stimulus-reward association task. Once the IT network was pretrained, its synaptic weights were fixed and did not change during the association task. The IT-dependent representation, $\mathbf{h_t}^{IT}$, for the current stimulus, $\mathbf{x_t}$, was derived as the output of the second encoder layer.

**IT and ET silencing.** We modeled IT chemogenetic silencing by replacing the IT-dependent input, $\mathbf{h}^{IT}$, to the value-encoding network with zero-mean isotropic Gaussian noise. As a result, the value-encoding network could no longer exploit the IT-dependent stimuli representations to estimate the value of each stimulus. Similarly, we reproduced ET chemogenetic silencing by replacing the value predictions relayed by ET neurons with isotropic Gaussian noise. As a result, RPEs could no longer be estimated correctly.

**IT representational overlap and learning performance.** For the IT representational overlap analysis, we used the eight Gaussian distributions of stimuli that pretrained the IT network. Randomly pairing these distributions allowed us to assess how overlap between IT representations of stimulus pairs (CS+ vs. CS−) influences learning in the stimulus-reward association task. We computed the mean Euclidean distance between IT representations, $\mathbf{h}^{IT}$, across paired stimuli and measured learning performance as the difference in stimulus association strength (i.e., value prediction) between CS+ and CS− after Session 2, where a larger positive difference indicated better learning.

## Statistics and reproducibility
No statistical method was used to predetermine sample size. Instead, the sample size was estimated based on the expected effect size based on similar studies (Takahashi et al., 2020; Benezra et al, 2024) and the current standard in mouse neuroscience studies. Poor data quality usually originates from movement artifacts during neuronal imaging of awake mice performing a behavior. Two-photon recordings with poor signal/data quality were excluded as assessed by visual inspection of the registered time-series images and registration metrics. Data exclusion was done prior to analysis, in order not to bias the exclusion criteria. Experiments were performed independently in different mice, and statistics were performed across mice in most cases. Stimuli presentations were randomized, and mice were randomly selected for the different experimental groups. Although the experimenter can tell during recording whether IT or ET neurons were recorded, all recorded neurons were analyzed in the same way, and the same statistical tests were performed in both groups. The same applies to the behavioral analysis of all the animal groups. Thus, the measurements are not affected by the lack of blinding.

## Reporting summary
Further information on research design is available in the Nature Portfolio Reporting Summary linked to this article.

## Data availability
The data used in this study are available via Zenodo at https://doi.org/10.5281/zenodo.17720603. Source data are provided with this paper.

## Code availability
The code used in this study are available via Zenodo at https://doi.org/10.5281/zenodo.17720603.

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

## Acknowledgements

We thank Andrada-Maria Marica for helpful discussions on modeling; Mario Carta and Richard Naud for their comments on an earlier version of the manuscript. This study was supported by the French National Centre for Scientific Research (to N.T.), the University of Bordeaux (2020 IdEx Junior Chair to N.T.), the Conseil Régional Nouvelle-Aquitaine (Bordeaux Neurocampus Junior Chair to N.T.), the ATIP-Avenir program (to N.T.), the Fondation Schlumberger pour l'Education et la Recherche (FSER202401018842 to N.T.), the Fondation pour la Recherche Médicale (EQU202403018077 to N.T.), the Agence Nationale de la Recherche (ANR-24-CE37-5761 and ANR-24-CE16-7349 to N.T.), the Brain Science Foundation (to N.T), the Research Foundation for Opto-Science and Technology (to N.T.), the EINSTEIN Foundation Berlin (PhD fellowship to S.M.; A-2021-644 to A.G.; EZ-2014-226 to D.S.), the Horizon Europe Research and Innovation Program (Marie Skłodowska-Curie Actions – 101148941 to C.M.), Deutsche Forschungsgemeinschaft (EXC-2049 – 390688087, LA 3442/3-1, LA 3442/6-1, 327654276/SFB1315 subprojects A04 & A10 to M.E.L.), the European Union Horizon 2020 Research and Innovation Programme (SGA1-3: 72070/HBP, 785907/HBP, 945539/HBP, 670118/ERC ActiveCortex, 101055340/ERC Cortical Coupling to M.E.L.), the Wellcome Trust (S122871-115 Transition Fellowship to M.G.),

the BBSRC (BB/X013340/1 to R.P.C.), EPSRC (EP/X029336/1 to R.P.C.), and the ERC-UKRI Frontier Research Guarantee Starting Grant (EP/Y027841/1 to R.P.C.). We thank the colleagues of the Research Workshop at the Charité - Universitätsmedizin Berlin for developing and manufacturing the experimental devices. We thank the colleagues of the viral vector core at the Charité - Universitätsmedizin Berlin for developing and manufacturing viral vectors used in this study. We acknowledge support by the Open Access Publication Fund of Humboldt-Universität zu Berlin.

## Author contributions

S.M., M.E.L., and N.T. conceived the project. S.M. and N.T. designed the experiments. S.M. and C.M. performed experiments. S.M., C.M., and N.T. performed the data analysis. M.G. and R.P.C. performed computational modeling. S.M. and N.T. wrote the paper with comments from the other authors. A.G. and D.S. provided feedback on the project.

## Funding

## Competing interests

The authors declare no competing interests.
