## [Transparent Peer Review file · Nature Communications]

Distinct roles of cortical layer 5 subtypes in associative learning

Corresponding Author: Dr Naoya Takahashi

Version 0:

Reviewer comments:

Reviewer #1

(Remarks to the Author)

The manuscript by Moberg et al. uses *in vivo* 2p imaging to examine the activity of layer 5 PNs in somatosensory cortex during learning and performance of a conditioned reward task. This work builds on earlier studies in both somatosensory and visual areas to explore heterogeneity of function among layer 5 subtypes -specifically ET and IT cells. Overall, the question is an interesting one, and this work certainly adds to our understanding of cortical cell type diversity. However, a number of issues reduced my enthusiasm for the manuscript in its present form.

1. The behavioral data should be characterized in much more detail. It is insufficient to present only AUC values without giving the actual Correct Rate and False-positive Rate values.

2. The neural data are very challenging to follow. First, the distinction between data in Figure 2 and Figure 3 (inside and outside the learning) are very hard to follow. As far as I can tell, these are the same animals, with data collected during a baseline (non-associative) period, during learning, and after learning (but where the stimuli and reward are again uncoupled). It would make far more sense to present these data together in a single figure, along a single time-axis, to understand how neuronal activity change during each stage.

Moreover, there are statements such as "responsiveness tended to decrease". But does this refer to fraction of cells? Why isn't the response amplitude similarly measured? Indeed, Figure 2 seems to only address fraction of responsive cells, while Figure 3 also shows DF/F values. Also, why are data in Figure 2 given as pie charts, while in Figure 3 similar data are in bar plots? These organizational choices make the manuscript very difficult to follow.

3. The argument that neural activity is not mostly driven by licking (Figure S2) are extremely unconvincing. It is well established that cortical activity leads motor activity in a variety of situations (see work from Churchland and Cardin Labs). The averaged data in Figure S2 simply suggest the same is true for S1 activity and licking. It would be far more compelling to show individual trials where the animal does not lick and demonstrate the S1 neural activity is unchanged.

4. Relatedly, the SVM data are difficult to interpret. If neural responses largely reflect licking, it is completely unsurprising that one can predict stimulus type (since CS+ evokes licking and CS- does not). Moreover, given the different stimulus-evoked activity for the two cell types, it is also unsurprising that the SVM performs differently for ET and IT. It is also surprising the authors did not attempt to predict behavioral performance rather than stimulus type. It would be far more powerful to use false-positive and error trials to tease out whether the neural activity is actual following the stimulus type or the behavior.

5. The interpretation of the chemogenetic data is unclear. Looking at Figure 5 and S5, it seems that ET silencing simply makes the animals lick more (increased anticipatory licking, increased false positive rate, decreased AUC). Again, more careful examination of correct/incorrect/false alarm trials would be helpful here.

6. The computational modeling is of little added value. Essentially, the model recreates experimental data but does not add any testable hypotheses to distinguish between alternative mechanistic possibilities. There may be an infinite set of models that can recapitulate the data - showing one of them does not add to our understanding.

7. The viral titer is not specified in the methods.

Reviewer #2

(Remarks to the Author)

In this manuscript, the authors present results from two-photon calcium imaging in subpopulations of S1 layer 5 neurons in a whisker stimulation trace conditioning paradigm. IT neurons stably encode stimulus identity and change very little with learning, whereas ET neuron responses to the CS stimuli and rewards undergo changes. While a large fraction of ET neurons already respond to rewards before learning, during the learning process, ET populations then increase their activity during the stimulus presentation of the CS+, reflecting the learned value (reward prediction) of the stimulus. A reinforcement learning model, combined with DREADD inhibition experiments, integrates the described findings of IT and ET neuron populations into a classical associative learning framework and creates predictions for how a circuit could use the characterized responses to form associations.

In summary, the study nicely identifies two circuit components that provide different information to an associative learning system. This approach of dissecting computations is very relevant, creates data that are valuable to the field and provides new entry points for future experiments to understand how brains learn stimulus-reward associations across distributed brain areas.

In its current state, the longitudinal imaging data is slightly underexplored, and since these chronic single neuron data are the strong suit of the presented study, it would be great to see a more detailed characterisation. Specifically, there seem to be two features to the change in ET neuron responses: 1) a shift earlier in time i.e. towards the reward predicting stimulus (from initial reward-related activity) and 2) also a selective increase in responses to CS+ over CS-. But the way the data are currently presented, it is hard to relate the two on the level of neuronal computations. By describing in more detail how response profiles of ET neurons change with learning (from initial reward responses to CS selectivity of individual neurons) a better understanding of the reward prediction calculation could be gained. Such an analysis could also be informed by doing a dropout analysis of the decoder (i.e. after learning, are any neurons particularly informative of the CS and, if so, how did their response profile change?).

Minor comments

Relating to muscimol inactivation of S1 (Fig 1f), it would be good to show not just auROC but also the lick rates to CS+ and CS- since there are two types of errors possible (indiscriminate licking to both CS or a lack of responsiveness to any CS). Relating to the passive sessions flanking the training, a more detailed description in the methods section/schematic is necessary (Fig 2c):

What was the block structure that is mentioned?

Was the lick spout present for the stimulus-only blocks, i.e. did the animal lick

Was the pre-training passive session the first time that the animals experienced those stimuli and or rewards in the setup? -> this will influence adaptation of neural responses and therefore the interpretation of the results in Fig 2

Is the sorting of neurons in Fig 4a and c different between the CS+ plots and the CS- plots? Along the lines of the main comment, it would be good to be able to visually relate how CS+ and CS- responses evolve in the same neurons (i.e. use the same sorting in an additional panel or compute CS selectivity directly and display that across days in heatmaps)

Reviewer #3

(Remarks to the Author)

In this article, Moberg et al. investigate the roles of cortical layer 5 subtypes in associative learning. By leveraging mouse genetics, calcium imaging, and neural manipulations, the authors come to the conclusion that intratelencephalic (IT; labeled with Tlx3-Cre) and extratelencephalic (ET; labeled with Sim1-Cre) layer 5 neurons in the barrel cortex exhibit distinct activity patterns during and play different roles in support of classical conditioning. Specifically, IT neurons have a relatively static representation of the CS+ and CS- stimuli over learning whereas ET neurons initially respond to rewards and then acquire a more robust response to the CS+ stimulus. Inhibition of either population throughout exposure to the task using Gi-coupled DREADDs disrupted learning with distinct effects on behavior. A simple computational reinforcement learning model incorporating key elements of the results recapitulates the dynamics of learning during Pavlovian conditioning.

Overall, the writing and data presentation is clear (with a few areas for improvement noted below), the key conclusions and claims are largely supported, and the work meets the standards in our field. I think this manuscript contributes important information to our understanding of how cortical cell types contribute to associative learning.

Major concerns:

Given findings that performance decreases after barrel cortex inactivation during the detection of tactile stimuli (e.g., O'Connor et al. 2010; Hong et al., 2018), and those from the authors' previous work indicates the detection threshold changes with inactivation of ET neurons (Takahashi et al., 2020), it is surprising that the muscimol injection in Figure 1g does not have any effect on performance in the present discrimination task. This dataset is small (N = 6 mice), variable, and likely underpowered. Increasing the sample size would lend more confidence to this finding.

On a related note, Figure S1 shows that the reversed contingency paradigm requires a much longer amount of training to

reach criterion performance. Does expert performance on the reversed paradigm require S1 activity? Answering this question would yield additional insight into when S1 is and is not required for associative learning and meaningfully contribute to this somewhat contentious subject. Furthermore, if S1 is not required, this result would increase confidence that the Figure 6 model applies generally to classical conditioning rather than just to the stimuli that the authors chose.

While the authors include a CNO-only test in wild-type animals (Figure S3), this control was not performed in the presence of a control viral vector (e.g., AAV2/1-hSyn-DIO-mCherry) injected into the C2 barrel cortex. It remains possible that an interaction between the CNO (and its metabolites at the 5 mg/kg dose), the surgical procedure, and viral expression could result in impaired learning. Doing this control in at least one of the Cre lines would strongly increase confidence in the DREADD-based silencing results.

Given the disproportionate numbers of males compared to females (45 vs. 9), there are likely many experiments that only include male animals. These experiments should be indicated throughout. It would be even better to include the numbers of male and female mice in each experiment. Without this information, it is difficult to determine whether both sexes are included in the key experiments that support the authors' conclusions.

Minor concerns:

Throughout (see Figure 1b for example), there appears to be a pause in licking when the water is dispensed on CS+ trials. I assume this is due to a technical issue preventing licks from being detected with the piezo-electric device when water is dispensed. If so, please clarify in legend or methods. If not, what is the reason for this pause?

Why are there different numbers of neurons in Figures 2f & j (between stimulus & reward graphs). I understand that the stimuli and reward were given in separate blocks, but were they not done in the same session? If so, this should be clarified. If not, why not? If I understand correctly, each of the three heatmaps were sorted separately. It would be useful to see the CS- heatmap sorted with index generated by the response to CS+ to more fully illustrate the extent to which the neurons that respond to the CS+ are the same as the neurons that respond to the CS- and the relative response magnitudes. Please describe intrinsic imaging and the analysis of intrinsic imaging data in more detail in methods section (e.g., what is the setup? what is the stimulus? what time periods were considered baseline and stimulus for analysis?).

Post hoc statistical tests should only be performed when the initial test reveals a significant result. Please check Figure S3 legend.

Because IT neurons and ET neurons are inactivated throughout the training period, the text in lines 12 through 16 on page 11 is confusing. Please rephrase for clarity.

Version 1:

Reviewer comments:

Reviewer #1

(Remarks to the Author)

The authors have done a nice job responding to my initial concerns. I think this work represents a valuable contribution to the field and have no further comments. I support publication.

Reviewer #2

(Remarks to the Author)

Thank you for addressing the points, overall the manuscript has improved considerably!

Especially the more detailed lick rate analyses and the consistent sorting of neuron heatmaps across time help a lot with the understanding of the paper.

In my opinion the manuscript is suitable for publication, but could still benefit from more detailed explanations/interpretations on the following questions:

Decoding analysis

How do you explain the difference in CS decoding performance between IT and ET populations at the naive stage (2h)? It looks like the proportion between unselective and selective neurons is comparable in both populations and the average response curves to CS+ and CS- look equally distinguishable (2e,g)?

How do you explain the stark difference between ET decoding at naive (chance level, 2h) and at day1 (significantly above chance, 3h)?

Response variability in ET neurons (esp with respect to model implementation)

How does the brain use their signal reliably if their response profile changes so considerably across days (did you address this somehow in the model?, 4f, S5)

How do you explain that the proportion of CS selective cells goes down with learning (3g)?

Reviewer #3

(Remarks to the Author)

The revised manuscript and reporting summary thoroughly addresses all my original concerns and recommend that it be accepted for publication.

Ref: NCOMMS-24-83466-T

=====**Point-by-Point Response to Reviewers**=====

Reviewer's comments are in italic.

Our answers are in blue.

Reviewer #1:

The manuscript by Moberg et al. uses in vivo 2p imaging to examine the activity of layer 5 PNs in somatosensory cortex during learning and performance of a conditioned reward task. This work builds on earlier studies in both somatosensory and visual areas to explore heterogeneity of function among layer 5 subtypes -specifically ET and IT cells. Overall, the question is an interesting one, and this work certainly adds to our understanding of cortical cell type diversity. However, a number of issues reduced my enthusiasm for the manuscript in its present form.

1. The behavioral data should be characterized in much more detail. It is insufficient to present only AUC values without giving the actual Correct Rate and False-positive Rate values.

We thank the review for this comment. In this study, we used a classical (Pavlovian) appetitive conditioning paradigm in which the conditioned response is measured as a continuous variable, i.e., the number of anticipatory licks to each CS. Because licking is not a binary outcome, categorical outcomes such as “correct” or “false-alarm” trials are less applicable. Following established practice (e.g., Otis et al., 2017; Namboodiri et al., 2019; Ruediger et al., 2020), we quantified differences in conditioned responses between CS+ and CS- using the area under the curve (AUC) of the licking behavior.

To provide a more comprehensive characterization of behavior, we have included lick rate plots for both CS+ and CS- conditions in Fig. 1c (the panel originally shown in Fig. 5f has been moved to Fig. 1c).

2. The neural data are very challenging to follow. First, the distinction between data in Figure 2 and Figure 3 (inside and outside the learning) are very hard to follow. As far as I can tell, these are the same animals, with data collected during a baseline (non-associative) period, during learning, and after learning (but where the stimuli and reward are again uncoupled). It would make far more sense to present these data together in a single figure, along a single time-axis, to understand how neuronal activity change during each stage.

Moreover, there are statements such as "responsiveness tended to decrease". But does this refer to fraction of cells? Why isn't the response amplitude similarly measured? Indeed, Figure 2 seems to only address fraction of responsive cells, while Figure 3 also shows DF/F values. Also, why are data in Figure 2 given as pie charts, while in Figure 3 similar data are in bar plots? These organizational choices make the manuscript very difficult to follow.

To improve clarity,

- We have reorganized the panels in Figs. 2 and 3 (e.g., moving the schematic of the imaging schedule during training from Fig. 2c to Fig. 3a) and revised the text to better explain our experimental design (Page 7, Line 17 in Main; Page 42, Line 10 in Methods).
- We decided to remove a comparison between before and after learning in order to focus on characterizing the responsiveness of IT and ET neurons in naive mice (before learning; Fig. 2) and during learning (Fig. 3).
- We have clarified in the text that "responsiveness" refers to the fraction of neurons that responded. To complement this, we have now added decoding analyses, which assess how reliably the population of IT or ET neurons encodes the stimulus (Fig. 2h).
- We have also revised the data presentation in Fig. 3: we replaced the bar plots with pie charts to ensure consistency with Fig. 2 and to facilitate comparison between the figures.

3. The argument that neural activity is not mostly driven by licking (Figure S2) are extremely unconvincing. It is well established that cortical activity leads motor activity in a variety of situations (see work from Churchland and Cardin Labs). The averaged data in Figure S2 simply suggest the same is true for S1 activity and licking. It would be far more compelling to show individual trials where the animal does not lick and demonstrate the S1 neural activity is unchanged.

In the original submission (Fig. S2, right panel) we included dF/F traces from trials in which mice did not show early lick responses. We observed that ET neurons were active even in the absence of licking, concluding "ET neuronal activity emerged with anticipatory licking but did not merely reflect lick-related motor activity" (original submission, Page 9, Line 9) and "this activity is at least partially independent of the motor response" (original submission, Page 18, Line 1). Our intent was not to suggest that neural activity is unrelated to licking, but rather to highlight additional contributions beyond motor output. That said, we agree with the reviewer that averaged traces alone are not fully persuasive. As the reviewer suggested, the revised manuscript now includes single-trial examples in

which ET responses do not align with lick onset (Fig. S3b). In addition, we performed two new analyses:

- Support vector regression (SVR) models trained solely on consummatory licks in CS+ trials failed to predict the number of anticipatory licks (Fig. S3a), indicating that ET neurons do not represent individual lick events.
- Neurons active during anticipatory licking were not necessarily the same as those recruited during consummatory licking after reward delivery (Fig. S3a, d).

These findings reinforce our original conclusion that ET activity is not simply a lick motor read-out. The results are discussed in the result section (Page 9, Line 16) and the discussion section (Page 17, Line 21). Moreover, our chemogenetic silencing of ET neurons in S1 prevented mice from learning to discriminate between the two CSs, indicating a causal contribution to learning beyond the encoding of licking behavior.

4. Relatedly, the SVM data are difficult to interpret. If neural responses largely reflect licking, it is completely unsurprising that one can predict stimulus type (since CS+ evokes licking and CS- does not). Moreover, given the different stimulus-evoked activity for the two cell types, it is also unsurprising that the SVM performs differently for ET and IT. It is also surprising the authors did not attempt to predict behavioral performance rather than stimulus type. It would be far more powerful to use false-positive and error trials to tease out whether the neural activity is actual following the stimulus type or the behavior.

We thank the reviewer for their insightful suggestion regarding the decoding analysis. Since the conditioned response in our task was measured as a continuous variable (i.e., anticipatory lick rate) rather than a categorical outcome, we performed SVR to assess whether ET neuronal activity predicts anticipatory licking behavior (Otis et al., 2017). Consistent with our observation that ET activity evolved in parallel with the development of anticipatory licking, SVR performance significantly improved as learning progressed (Fig. 3i). Importantly, ET activity did not simply reflect lick counts (Fig. S3; also see our response to Comment #3), suggesting that it at least partially represents the decision to lick or reward expectation, rather than a direct lick motor command. Based on these results, we have revised our interpretation of the SVM results decoding CS identity as follows: “we interpret the improved SVM decoding of CS identity during learning not as a consequence of enhanced sensory responsiveness of ET neurons, which is weak in naïve mice (Fig. 2f–h), but rather as a readout of emerging behavioral performance through reward expectation signals.” (Page 10. Line 3)

5. The interpretation of the chemogenetic data is unclear. Looking at Figure 5 and S5, it seems that ET silencing simply makes the animals lick more (increased anticipatory licking, increased false positive rate, decreased AUC). Again, more careful examination of correct/incorrect/false alarm trials would be helpful here.

It is correct that silencing ET neurons increased licking to both CS+ and CS- trials during learning (Fig. 5d, f, S6b, c), resulting in reduced AUC (Fig. 5e). We interpret this as indicating that ET silencing prevents mice from learning to differentiate between CS+ and CS-, prompting mice to lick indiscriminately to both CSs. In contrast, in expert mice with established responses, ET silencing did not alter the learned pattern of licking (i.e., increased anticipatory licking to CS+ and reduced licking to CS-; Fig. S6f). Thus, the indiscriminate licking observed during training is not a general effect of ET silencing, but reflects a specific impairment of the learning mechanism needed to differentiate between conditioned stimuli. We clarified this point in the result section (Page 12, Line 3).

6. The computational modeling is of little added value. Essentially, the model recreates experimental data but does not add any testable hypotheses to distinguish between alternative mechanistic possibilities. There may be an infinite set of models that can recapitulate the data - showing one of them does not add to our understanding.

We appreciate the reviewer's critical feedback regarding the value of our computational model. We agree that a model's true utility lies in its ability to generate testable hypotheses that provide insight into underlying mechanisms. Our circuit model is not intended as a mere recapitulation of the experimental data, but rather as a mechanistic proposal that integrates our empirical findings within the classical reinforcement learning framework.

The core contribution of our model is the explicit division of labor hypothesis it proposes: that L5 IT neurons contribute to value estimation, while L5 ET neurons are crucial for transmitting signals to compute reward prediction errors (RPEs), specifically negative RPEs. This specific functional distinction, which is a novel conceptualization, is directly testable.

As we now clarify in the revised manuscript (Page 15, Line 25), a key prediction of our model is that directly manipulating CS representations encoded by IT neurons would specifically alter the reward-predicting signal in ET neurons. Given our established ability to selectively modulate these neurons, this provides a feasible path for future research to directly test this prediction. Therefore, our model's specific assumptions, that one subpopulation handles sensory coding for value estimation and the other reward-expectation signaling for computing RPEs, provide a predictive framework for understanding the distinct contributions of IT and ET neurons to learning.

7. The viral titer is not specified in the methods.

We have added the titer information in the method section.

Reviewer #2:

In this manuscript, the authors present results from two-photon calcium imaging in subpopulations of S1 layer 5 neurons in a whisker stimulation trace conditioning paradigm. IT neurons stably encode stimulus identity and change very little with learning, whereas ET neuron responses to the CS stimuli and rewards undergo changes. While a large fraction of ET neurons already respond to rewards before learning, during the learning process, ET populations then increase their activity during the stimulus presentation of the CS+, reflecting the learned value (reward prediction) of the stimulus. A reinforcement learning model, combined with DREADD inhibition experiments, integrates the described findings of IT and ET neuron populations into a classical associative learning framework and creates predictions for how a circuit could use the characterized responses to form associations.

In summary, the study nicely identifies two circuit components that provide different information to an associative learning system. This approach of dissecting computations is very relevant, creates data that are valuable to the field and provides new entry points for future experiments to understand how brains learn stimulus-reward associations across distributed brain areas.

In its current state, the longitudinal imaging data is slightly underexplored, and since these chronic single neuron data are the strong suit of the presented study, it would be great to see a more detailed characterisation. Specifically, there seem to be two features to the change in ET neuron responses: 1) a shift earlier in time i.e. towards the reward predicting stimulus (from initial reward-related activity) and 2) also a selective increase in responses to CS+ over CS-. But the way the data are currently presented, it is hard to relate the two on the level of neuronal computations. By describing in more detail how response profiles of ET neurons change with learning (from initial reward responses to CS selectivity of individual neurons) a better understanding of the reward prediction calculation could be gained. Such an analysis could also be informed by doing a dropout analysis of the decoder (i.e. after learning, are any neurons particularly informative of the CS and, if so, how did their response profile change?).

We thank the reviewer for pointing out that the longitudinal imaging data were underexplored. We performed additional analyses to more fully leverage the power of across-day tracking. In the revised manuscript, we now include:

- A Sankey diagram illustrating how CS responsiveness of individual neurons changes over days, providing a quantitative overview of neuronal dynamics across learning (Fig. 4b, f).
- An analysis of changes in CS selectivity at the single-cell level, comparing individual neurons at the beginning and end of learning (Fig. 4c, g).

- Across-day activity profiles of neurons initially responsive to reward, showing how their responses evolve with training (Fig. S5).

These additions clarify how neuronal coding evolves over time. We revised the main text accordingly.

Minor comments

Relating to muscimol inactivation of S1 (Fig 1f), it would be good to show not just auROC but also the lick rates to CS+ and CS- since there are two types of errors possible (indiscriminate licking to both CS or a lack of responsiveness to any CS).

Thank you for the comment. We have now included lick rate plots for the muscimol experiment (Fig. 1g), which shows limited responsiveness to any CS during learning, similar to the pattern we observed with IT neuronal inactivation in Fig. 5c.

Relating to the passive sessions flanking the training, a more detailed description in the methods section/schematic is necessary (Fig 2c):

What was the block structure that is mentioned?

Thank you for spotting this. We reorganized the panels in Figs. 2 and 3, and revised the text to better explain our experimental design (Page 42, Line 10).

Was the lick spout present for the stimulus-only blocks, i.e. did the animal lick

In stimulus-only sessions, where stimulus was given without any rewards, the lick spout was present to record any licking of the mouse. However, we excluded trials in which the animal licked when analyzing the neuronal activity to exclude any potential confounds of licking on the neuronal activity. We have added the following paragraph in the method section: "To isolate pure stimulus responses and exclude any potential confound of licking, we excluded trials during the stimulus-only blocks in which the mice licked (IT neuronal recording: 0.17 ± 0.41 out of 25 trials; ET neuronal recording: 0.83 ± 1.17 out of 25 trials)." (Page 44, Line 22).

Was the pre-training passive session the first time that the animals experienced those stimuli and or rewards in the setup? -> this will influence adaptation of neural responses and therefore the interpretation of the results in Fig 2

During the pre-training stimulus-only sessions, this was the first time the animals were exposed to the two stimuli. However, the mice had been habituated to drinking water from

the lick spout for a few days prior to the start of the experiments, thus the lick spout was not novel to them.

Is the sorting of neurons in Fig 4a and c different between the CS+ plots and the CS- plots? Along the lines of the main comment, it would be good to be able to visually relate how CS+ and CS- responses evolve in the same neurons (i.e. use the same sorting in an additional panel or compute CS selectivity directly and display that across days in heatmaps)

Yes, the sorting orders were different — in the original submission, neurons in the CS+ plots were sorted based on their response amplitudes to CS+, while neurons in the CS- plots were sorted based on their responses to CS-. We have clarified this point in the revised text. Following the reviewer's suggestion, we have now added new activity heatmaps for CS+ and CS- trials, in which neurons are sorted in the same order (Fig. S4). In addition, we have included a new plot quantifying the CS selectivity of individual neurons, comparing these values between Day 1 and Day 5 (Fig. 4c, g).

Reviewer #3:

In this article, Moberg et al. investigate the roles of cortical layer 5 subtypes in associative learning. By leveraging mouse genetics, calcium imaging, and neural manipulations, the authors come to the conclusion that intratelencephalic (IT; labeled with Tlx3-Cre) and extratelencephalic (ET; labeled with Sim1-Cre) layer 5 neurons in the barrel cortex exhibit distinct activity patterns during and play different roles in support of classical conditioning. Specifically, IT neurons have a relatively static representation of the CS+ and CS- stimuli over learning whereas ET neurons initially respond to rewards and then acquire a more robust response to the CS+ stimulus. Inhibition of either population throughout exposure to the task using Gi-coupled DREADDs disrupted learning with distinct effects on behavior. A simple computational reinforcement learning model incorporating key elements of the results recapitulates the dynamics of learning during Pavlovian conditioning.

Overall, the writing and data presentation is clear (with a few areas for improvement noted below), the key conclusions and claims are largely supported, and the work meets the standards in our field. I think this manuscript contributes important information to our understanding of how cortical cell types contribute to associative learning.

Major concerns:

Given findings that performance decreases after barrel cortex inactivation during the detection of tactile stimuli (e.g., O'Connor et al. 2010; Hong et al., 2018), and those from the authors' previous work indicates the detection threshold changes with inactivation of ET neurons (Takahashi et al., 2020), it is surprising that the muscimol injection in Figure 1g does not have any effect on performance in the present discrimination task. This dataset is small (N = 6 mice), variable, and likely underpowered. Increasing the sample size would lend more confidence to this finding.

In our previous studies (Takahashi et al., 2016; 2020), we demonstrated that detection of near-threshold whisker stimuli (~1-2° deflections) relies on S1 activity. However, we also found that manipulations of S1 had little effect on detection when the stimuli were twice the detection threshold. In the present study, we intentionally used much stronger whisker stimuli (~10-20°) — an order of magnitude larger than threshold stimuli — to ensure that the stimuli were well above the animals' perceptual threshold. Consistent with our earlier findings, S1 inactivation did not affect conditioned responses to these high-intensity stimuli in expert mice. A similar result was reported in V1 (Ruediger et al., 2020). We now clarify this point in the method section (Page 41, Line 20).

Nevertheless, in response to the reviewer's concern about sample size, we performed additional muscimol inactivation experiments, increasing the sample size to $n = 9$ mice to improve statistical power (Fig. 1i). These additional data confirmed our original conclusion that performance in expert mice remained unaffected by S1 inactivation.

On a related note, Figure S1 shows that the reversed contingency paradigm requires a much longer amount of training to reach criterion performance. Does expert performance on the reversed paradigm require S1 activity? Answering this question would yield additional insight into when S1 is and is not required for associative learning and meaningfully contribute to this somewhat contentious subject. Furthermore, if S1 is not required, this result would increase confidence that the Figure 6 model applies generally to classical conditioning rather than just to the stimuli that the authors chose.

We thank the reviewer for suggesting this experiment. We have now performed muscimol inactivation experiments in expert mice trained on the reversed paradigm. Consistent with the results from the original paradigm, silencing S1 did not affect performance in expert animals, supporting the general applicability of our findings and model to classical conditioning beyond the specific stimuli used. These new data have been added in Fig. S1c.

While the authors include a CNO-only test in wild-type animals (Figure S3), this control was not performed in the presence of a control viral vector (e.g., AAV2/1-hSyn-DIO-mCherry) injected into the C2 barrel cortex. It remains possible that an interaction

between the CNO (and its metabolites at the 5 mg/kg dose), the surgical procedure, and viral expression could result in impaired learning. Doing this control in at least one of the Cre lines would strongly increase confidence in the DREADD-based silencing results.

To address this concern, we performed additional experiments in which we tested the effect of CNO alone in mice injected with a control viral vector (AAV9-EF1a-DIO-mCherry) into the C2 barrel cortex. These experiments were conducted in six Tlx3-Cre mice. The results confirmed that CNO administration had no effect on learning in these control animals, supporting the conclusion that the behavioral effects observed in our DREADD experiments are not due to off-target effects of CNO or interactions with surgery or viral expression. These new data have been added in Fig. S6a.

Given the disproportionate numbers of males compared to females (45 vs. 9), there are likely many experiments that only include male animals. These experiments should be indicated throughout. It would be even better to include the numbers of male and female mice in each experiment. Without this information, it is difficult to determine whether both sexes are included in the key experiments that support the authors' conclusions.

We have included this information in the section “Reporting for specific materials, systems and methods” in the “Reporting Summary” file.

Minor concerns:

Throughout (see Figure 1b for example), there appears to be a pause in licking when the water is dispensed on CS+ trials. I assume this is due to a technical issue preventing licks from being detected with the piezo-electric device when water is dispensed. If so, please clarify in legend or methods. If not, what is the reason for this pause?

We thank the reviewer for this observation. There is indeed a drop in lick frequency immediately after the water droplet was given. We have clarified this in the method section as follows: “During CS+ trials, we frequently observed a brief pause in licking upon water delivery. This interruption may arise from the behavioral shift between anticipatory and consummatory licking, or from a technical limitation: the animal could lick the dispensed droplet without touching the spout, causing the piezo sensor to miss those licks.” (Page 42, Line 3).

Why are there different numbers of neurons in Figures 2f & j (between stimulus & reward graphs). I understand that the stimuli and reward were given in separate blocks, but were they not done in the same session? If so, this should be clarified. If not, why not?

The data from Fig. 2f, j are from two different blocks that were not necessarily from the same sessions, thus the number of neurons differ as well. We have revised Fig. 2 and clarified the experimental procedure in the method section (Page 42, Line 10).

If I understand correctly, each of the three heatmaps were sorted separately. It would be useful to see the CS- heatmap sorted with index generated by the response to CS+ to more fully illustrate the extent to which the neurons that respond to the CS+ are the same as the neurons that respond to the CS- and the relative response magnitudes.

We have now added new heatmaps sorted with index generated by the response to CS+ (Fig. S4).

Please describe intrinsic imaging and the analysis of intrinsic imaging data in more detail in methods section (e.g., what is the setup? what is the stimulus? what time periods were considered baseline and stimulus for analysis?).

Details of intrinsic imaging have been elaborated in the method section (Page 40, Line 14).

Post hoc statistical tests should only be performed when the initial test reveals a significant result. Please check Figure S3 legend.

Thank you for spotting this. We have removed the post-hoc statistical tests from the concerning figure.

Because IT neurons and ET neurons are inactivated throughout the training period, the text in lines 12 through 16 on page 11 is confusing. Please rephrase for clarity.

We have changed the text as follows: “Closer inspection of anticipatory licking revealed divergent effects of chronic inactivation during training (Fig. 5f).” (Page 11, Line 17)

References

1. Otis, J.M., Nambodiri, V.M., Matan, A.M., Voets, E.S., Mohorn, E.P., Kosyk, O., McHenry, J.A., Robinson, J.E., Resendez, S.L., Rossi, M.A. & Stuber, G.D. Prefrontal cortex output circuits guide reward seeking through divergent cue encoding. *Nature* **543**, 103-107 (2017).
2. Nambodiri, V.M.K., Otis, J.M., van Heeswijk, K., Voets, E.S., Alghorazi, R.A., Rodriguez-Romaguera, J., Mihalas, S. & Stuber, G.D. Single-cell activity tracking

reveals that orbitofrontal neurons acquire and maintain a long-term memory to guide behavioral adaptation. *Nat Neurosci* **22**, 1110-1121 (2019).

3. Ruediger, S. & Scanziani, M. Learning speed and detection sensitivity controlled by distinct cortico-fugal neurons in visual cortex. *eLife* **9**:e59247 (2020).
4. Takahashi, N., Oertner, T.G., Hegemann, P. & Larkum, M.E. Active cortical dendrites modulate perception. *Science* **354**, 1587-1590 (2016).
5. Takahashi, N., Ebner, C., Sigl-Glockner, J., Moberg, S., Nierwetberg, S. & Larkum, M.E. Active dendritic currents gate descending cortical outputs in perception. *Nat Neurosci* **23**, 1277-1285 (2020).

Ref: NCOMMS-24-83466-T

=====**Point-by-Point Response to Reviewers**=====

Reviewer's comments are in italic.

Our answers are in blue.

Reviewer #1:

The authors have done a nice job responding to my initial concerns. I think this work represents a valuable contribution to the field and have no further comments. I support publication.

We thank the reviewer for their valuable and constructive comments during the previous revision.

Reviewer #2:

Thank you for addressing the points, overall the manuscript has improved considerably! Especially the more detailed lick rate analyses and the consistent sorting of neuron heatmaps across time help a lot with the understanding of the paper. In my opinion the manuscript is suitable for publication, but could still benefit from more detailed explanations/interpretations on the following questions:

We thank the reviewer for the positive assessment of our previously revised manuscript and for the additional questions. We have further revised the text to address these points in the new version of the manuscript.

Decoding analysis

How do you explain the difference in CS decoding performance between IT and ET populations at the naive stage (2h)? It looks like the proportion between unselective and selective neurons is comparable in both populations and the average response curves to CS+ and CS- look equally distinguishable (2e,g)?

In Fig. 2e,g, the proportion of selective neurons and their response profiles are quantified by averaging responses across trials within the session. This averaging makes IT and ET populations appear similarly selective at the naive stage. However, the decoding analysis operates on single trials, assessing how well population activity can predict stimulus identity on a trial-by-trial basis. This measure depends strongly on the reliability and consistency of neuronal responses across trials, not just on the mean response profile. Thus, the difference in decoding performance indicates that, in naive mice, IT neurons respond to CSs more reliably across trials, whereas ET neurons show sparser and less consistent responses, even though both populations exhibit comparable average selectivity when responses are trial-averaged.

To clarify this point, we have added the following sentence in the result section.
“The finding that some ET neurons show stimulus selectivity while population decoding remains poor indicates that ET population activity is sparse and unreliable at the single-trial level. (Page 8, Line 8)”

How do you explain the stark difference between ET decoding at naive (chance level, 2h) and at day1 (significantly above chance, 3h)?

On Day 1, animals already begin to discriminate CS+ from CS- (Fig. 1c-e, 3a). As ET neurons develop reward-expectation signals, their responses become more differentiated between CSs already on Day 1, leading to a sharp increase in decoding performance compared to the naive stage.

To clarify this point, we have added the following sentence in the result section.
“This improvement was already evident relative to naïve mice (Fig. 2h), consistent with the animals beginning to acquire the task on Day 1 (Fig. 3a). (Page 9, Line 18)”

*Response variability in ET neurons (esp with respect to model implementation)
How does the brain use their signal reliably if their response profile changes so considerably across days (did you address this somehow in the model?, 4f, S5)*

Thank you for raising this important point. Although individual ET neurons show day-to-day variability in their response profiles, the population-level reward-expectation signal remains stable. We hypothesize that downstream circuits likely rely on the collective ET activity, not on which specific ET neurons carry the signal. Thus, even if the contributing neurons change over days, the summed population output that encodes reward expectation is preserved. This principle is also reflected in our model, which reads out population activity, not the identity of individual ET neurons.

We have added the following paragraph in the discussion section.
“Although ET responses gradually stabilized across training, individual neurons showed substantial day-to-day variability in their response profiles. This variability suggests that downstream circuits likely rely on the collective activity of the ET population rather than on specific neurons. In this view, even if the subset of responsive ET neurons shifts across days, the population-level signal representing reward expectation remains stable. Our model is consistent with this principle. Determining how downstream targets integrate this population signal will be an important direction for future work. (Page 19, Line 1)”

How do you explain that the proportion of CS selective cells goes down with learning (3g)?

Related to our response to Comment #1, the proportion of CS-selective neurons is based on trial-averaged responses, which does not capture the trial-to-trial reliability of each ET neuron. Although the averaged selectivity appears to decrease, the decoding analysis shows a monotonic increase in ET population accuracy across learning (Fig. 3h). This indicates that, on a single-trial basis, ET population activity actually becomes more

selective, even if this is not reflected in the proportion of selective cells defined by averaged responses.

Reviewer #3:

The revised manuscript and reporting summary thoroughly addresses all my original concerns and recommend that it be accepted for publication.

We thank the reviewer for their valuable and constructive comments during the previous revision.